# Discovery of a non-canonical prototype long-chain monoacylglycerol lipase through a structure-based endogenous reaction intermediate complex

Nikos Pinotsis [1,5,6], Anna Krüger [2,6], Nicolas Tomas[3], Spyros D. Chatziefthymiou[1], Claudia Litz[1], Simon Arnold Mortensen [1], Mamadou Daffé[3], Hedia Marrakchi[3], Garabed Antranikian[2] & Matthias Wilmanns [1,4] ✉

The identification and characterization of enzyme function is largely lacking behind the rapidly increasing availability of large numbers of sequences and associated high-resolution structures. This is often hampered by lack of knowledge on in vivo relevant substrates. Here, we present a case study of a high-resolution structure of an unusual orphan lipase in complex with an endogenous C18 monoacylglycerol ester reaction intermediate from the expression host, which is insoluble under aqueous conditions and thus not accessible for studies in solution. The data allowed its functional characterization as a prototypic long-chain monoacylglycerol lipase, which uses a minimal lid domain to position the substrate through a hydrophobic tunnel directly to the enzyme's active site. Knowledge about the molecular details of the substrate binding site allowed us to modulate the enzymatic activity by adjusting protein/substrate interactions, demonstrating the potential of our findings for future biotechnology applications.

The annotation of relevant enzyme function under physiological conditions remains a major challenge, despite the availability of millions of protein sequences and experimentally determined or predicted molecular structures associated with them[1,2]. Enzymes often have different and broader substrate profiles within their natural cellular environment than experimentally accessible under in vitro conditions or by using purified protein samples. Hence, albeit powerful tools have been developed to predict enzymatic function, options to prove them in vivo remain limited[3–8]. High-resolution structural biology allows the direct visualization of enzyme-ligand complexes at atomic detail and thus presents a highly attractive tool for enzyme function discovery, but it requires the availability of pure compounds for structural analysis. Although protein structures in the presence of large hydrophobic substrates or ligands have been published, often with a requirement of applying target-tailored solubilization protocols[9–12], they still remain exceptional and hence available structure repositories remain mostly populated with protein structures of soluble, small ligands. Alternatively, enzyme structures could also be determined in the presence of endogenous substrates or catalysis intermediates taken up from their cellular environment for direct

[1]European Molecular Biology Laboratory, Hamburg Unit, Notkestrasse 85, 22607 Hamburg, Germany. [2]Hamburg University of Technology, Kasernenstrasse 12, 21073 Hamburg, Germany. [3]Institut de Pharmacologie et de Biologie Structurale, Université de Toulouse, CNRS, Université Toulouse III-Paul Sabatier, Toulouse, France. [4]University Hamburg Clinical Center Hamburg-Eppendorf, Martinistrasse 52, 20251 Hamburg, Germany. [5]Present address: Department of Chemistry, National and Kapodistrian University of Athens, Zografou, Greece. [6]These authors contributed equally: Nikos Pinotsis, Anna Krüger. ✉e-mail: matthias.wilmanns@embl-hamburg.de

evidence of enzyme-substrate relationships, but successful instances have remained rare to date[13]. This requires conditions where substrate turnover is inhibited by genetic modification of enzyme targets or experimental conditions that lead to lack of catalytic activity. In this study, we demonstrate the power of an endogenous substrate uptake for structural and subsequent functional characterization of an orphan enzyme with putative lipase/esterase activity, which otherwise would not have been accessible due to lack of solubility.

Lipases are enzymes that reversibly catalyze the hydrolysis and synthesis of a large variety of glycerol esters[14–16]. Knowledge about these enzymes has allowed the development of numerous lipase-catalyzed processes in medical biotechnology, detergent industry, organic synthesis, biodiesel production, agrochemical industry, flavor and aroma industry, and food production[17–19]. In contrast to many other carboxyl ester hydrolases, lipases are active mainly at water-lipid interfaces, a process known as interfacial activation[20,21]. In essence, long-chain glycerol esters are delivered in non-monomeric emulsions rather than water-soluble substrates. Interfacial activation of lipases is thought to require an additional α-helical domain referred to as "lid", alternatively also named "cap", which contains several hydrophilic residues in the vicinity of the active site[22]. In contrast to the catalytic domain (CD), there is only little sequence conservation of the lid. Available structural data on lipases show highly diverse lid topology and structure, suggesting variable mechanisms to promote active site opening for access of acyl substrates and rendering precise prediction of enzymatic substrate specificity challenging[12,23,24]. Lipases from extremophilic microorganisms are of superior interest for applications in industrial processes since demanding conditions such as high temperature or high salt concentration do not impair their activity[25–27].

Most of the previous molecular activity studies have been limited to isolated or recombinant lipases. Given that quantitative lipase activity measurements are generally hampered by the limited solubility of long-chain glycerol esters under those experimental conditions it is plausible to assume that the substrate profile differs under cellular conditions.

Based on turnover measurements with a broad spectrum of substrates, an enzyme from the thermophilic anaerobic bacterium *Thermoanaerobacter thermohydrosulfuricus (Tth)* was tentatively assigned as a lipase[28]. This strain was originally isolated from Solar Lake located on the Sinai peninsula, which has an extreme marine environment with a temperature range from 16 to 60 °C and high level of salinity. These environmental conditions have given rise to complex biochemical phenomena that are linked to cycles of evaporation and infiltration of water external sources. Isolates from Solar Lake have shown remarkable biochemical processes related to the degradation of starch, amylose, and pullulan[29,30]. As this enzyme revealed a remarkable level of robustness against a variety of solvents and temperatures, it represents an attractive target for potential biotechnology applications. However, underlying molecular mechanisms explaining the enzyme's activity properties and substrate promiscuity remained unknown.

In this work, we determined the structure of the enzyme revealing a long-chain C18 monoacylglycerol (MAG) bound to the active site, which was taken up from its expression host. Its distal part is embedded by a tunnel formed by an unconventional lid domain with minimal α-helix-β-hairpin-α-helix (HBH) topology. In cell extracts that overexpress the enzyme, we found high levels of C16/18 fatty acid (FA) products, in agreement with our structural data, demonstrating that the *Tth* enzyme functions as a long-chain MAG lipase. *Tth* MAG lipase prototypes a new lipase/esterase family with HBH lid domains comprising sequences inserted into the HBH β-hairpin that are highly divergent in length and structure. Our findings demonstrate an example of using high-resolution structural biology where complex formation with an endogenous reaction intermediate has become achievable to elucidate its function.

## Results and discussion

### *Tth* MAG lipase reveals an α/β hydrolase fold with a minimal HBH lid domain

To provide insight into the *Tth* MAG lipase active-site topography and shed light on its unique substrate promiscuity, we characterized the enzyme structurally and functionally. Purified *Tth* MAG lipase elutes as a dimer (Fig. 1a). Its fold melting temperature measured by nano differential scanning fluorimetry (nanoDSF) is at 77 °C, which is within expectation in light of the thermophilic origin of *Tth* (Fig. 1b, Supplementary Table 1). Next, we determined its high-resolution crystal structure, both in the presence and absence of the covalent active site serine protease inhibitor phenylmethylsulfonyl fluoride (PMSF) (Figs. 1c, e, 2, Supplementary Table 2). The overall fold of dimeric *Tth* MAG lipase can also be reliably modeled using the AlphaFold2 and MMseqs2 package (Supplementary Fig. 1)[31,32]. However, as there are significant differences in the areas of the bound ligands and the dimeric interface, in the remaining contribution we refer to our experimental structures only.

Both *Tth* MAG lipase structures—the active and phenylmethylsulfonyl (PMS)-bound—reveal a two-fold symmetric dimer, in agreement with its dimeric assembly in solution (Fig. 1a, e). The *Tth* MAG lipase dimer is formed by an almost flat interface area which represents about 10 % of the total solvent accessible area (Supplementary Fig. 2). The interface is established through an intricate network of charged interactions involving residues from both domains, with the majority of the interactions originating from residues of the CD. It forms a large cleft that connects the two active sites of the two protomers (Fig. 1e). Converting the charge of one of the most central interface residues (E72R) renders a monomeric version of the protein, confirming that the interface observed in the structure is relevant (Fig. 1a).

A minimal lid domain comprising the residues 141-183 is inserted between strand β6 and helix αE on top of the CD active-site area (Fig. 1c–e, Supplementary Fig. 2). Different from most other lipases, in which the lid is generally formed by a bundle of α-helices[22,24], in *Tth* MAG lipase this domain is compact. It is formed by two small α-helices, followed by a β-hairpin and an additional α-helix, referred to as α-helix/β-hairpin/α-helix (HBH) (Fig. 1c, Supplementary Fig. 2). In both *Tth* MAG lipase structures, the orientation of the HBH lid to the CD is basically identical (Fig. 2d, e), dismissing any large-scale opening/closing mechanisms previously shown for other lid domain-containing lipases[22,24]. Despite the presence of the lid domain, the MAG lipase active site remains largely accessible (Fig. 1e).

### Long chain MAG reaction intermediate is held by a hydrophobic lid tunnel

In the active site of both *Tth* MAG lipase structures, with and without PMSF, we observed literally identical density of a glycerol molecule, presenting one of the two final products of esterase/lipase catalysis (Figs. 2a, b, 3, Supplementary Fig. 3a). The glycerol molecule tightly interacts via its two C2 and C3 hydroxyl groups with the carboxylate group of a non-conserved active-site residue Glu43, specific to *Tth* MAG lipase (Fig. 2a, b, Supplementary Fig. 5a). As di- and triacylglycerol (DAG, TAG) esters would require linkages through these two hydroxyl groups, only MAG ester binding via the C1 hydroxyl group of the glycerol backbone is sterically possible. The dihedral conformation of this residue is in an energetically less favored area of the Ramachandran plot in both *Tth* MAG lipase structures (Supplementary Fig. 6), suggesting that its role in glycerol binding may supersede any other lower energy conformation.

In both protomers of the active *Tth* MAG lipase structure, we found additional extended ligand electron density pointing away from the active site crossing through a 15 Å long tunnel formed by the HBH lid domain (Supplementary Fig. 3a). As we did not add any substrate to the crystallization buffer, bound ligands can only originate from the

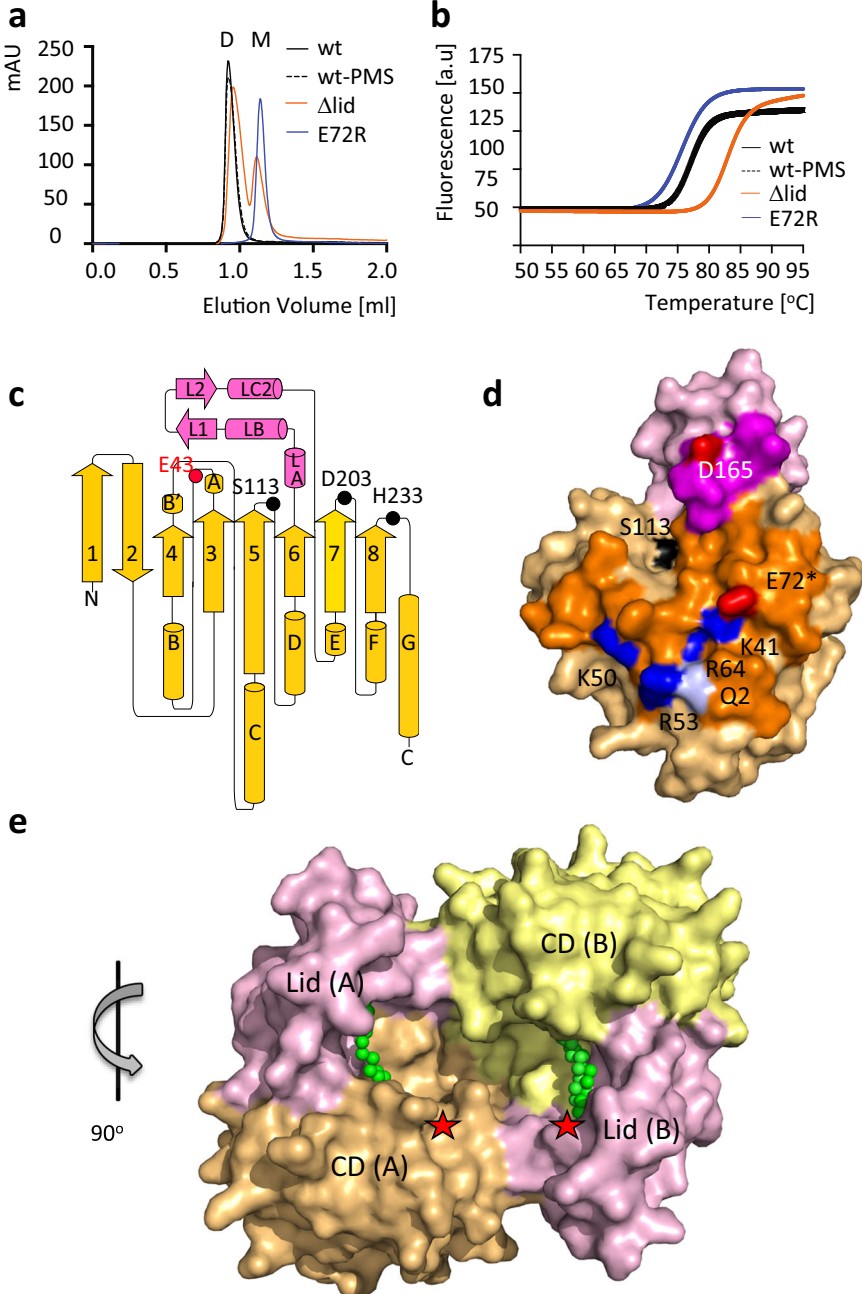

**Fig. 1 | Biophysical and structural properties of *Tth* MAG lipase. a** Size exclusion chromatograms indicating dimeric (D) and monomeric (M) fractions and **b** nano differential scanning fluorimetry profiles of selected *Tth* MAG lipase variants. For further analysis, see Supplementary Table 1. **c** topology diagram of the molecular structure of the *Tth* MAG lipase protomer. CD, yellow; lid domain, pink. Secondary structural elements are in proportion to their length and are labeled. The positions of key active site residues (cf. Supplementary Fig. 5) are highlighted. The active site glycerol interacting residue Glu43 is shown in red. While in the text three-letter codes were used for residue specifications, for the reason of clarity one-letter codes were used in all Figures. **d** surface presentation of *Tth* MAG lipase protomer, with the dimeric interface surface highlighted in stronger colors. Polar residues that contribute to specific interactions to the dimeric interface are shown in red

(glutamate, aspartate), blue (arginine, lysine), and pale blue (glutamine), and are labeled. Glu72, which was mutated into an arginine for probing the dimeric interface, is highlighted by an asterisk. The surface of the active site residue Ser113 is shown in black, as point of reference for locating the active site. **e** *Tth* MAG lipase dimer in surface presentation. The catalytic domains (CD) and lid domains of each protomer are colored separately. The two *Tth* MAG lipase protomers form a large joint substrate binding site, as indicated by the bound C18 MAG ligands (Fig. 2a) and two asterisks in red, indicating the two active sites. In this presentation, the proximal part of the C18 MAG ligands is visible (cf. Fig. 2). The orientation of *Tth* MAG lipase protomer A is rotated by about 90 degrees around a vertical axis with respect to the *Tth* MAG lipase monomer shown in panel (**d**), as indicated. Source data are provided as a Source Data file.

*Escherichia* (*E.*) *coli* expression host FA pool. To rule out that the presence of glycerol in the *Tth* MAG lipase structure is due to the addition of glycerol in the cryo-protectant buffer, we determined an additional *Tth* MAG lipase structure without using glycerol in any sample preparation step and cryo-protection prior to X-ray data collection (Supplementary Table 2). In this structure, glycerol is found in a literally

identical active site position in five out of six protein chains of the respective asymmetric unit, confirming its origin to be independent from a specific crystallization and cryo-protection protocol (Supplementary Fig. 7). To independently identify the ligand content of the extended electron density, we analyzed the crystals of the active enzyme by reversed phase liquid chromatography followed by mass

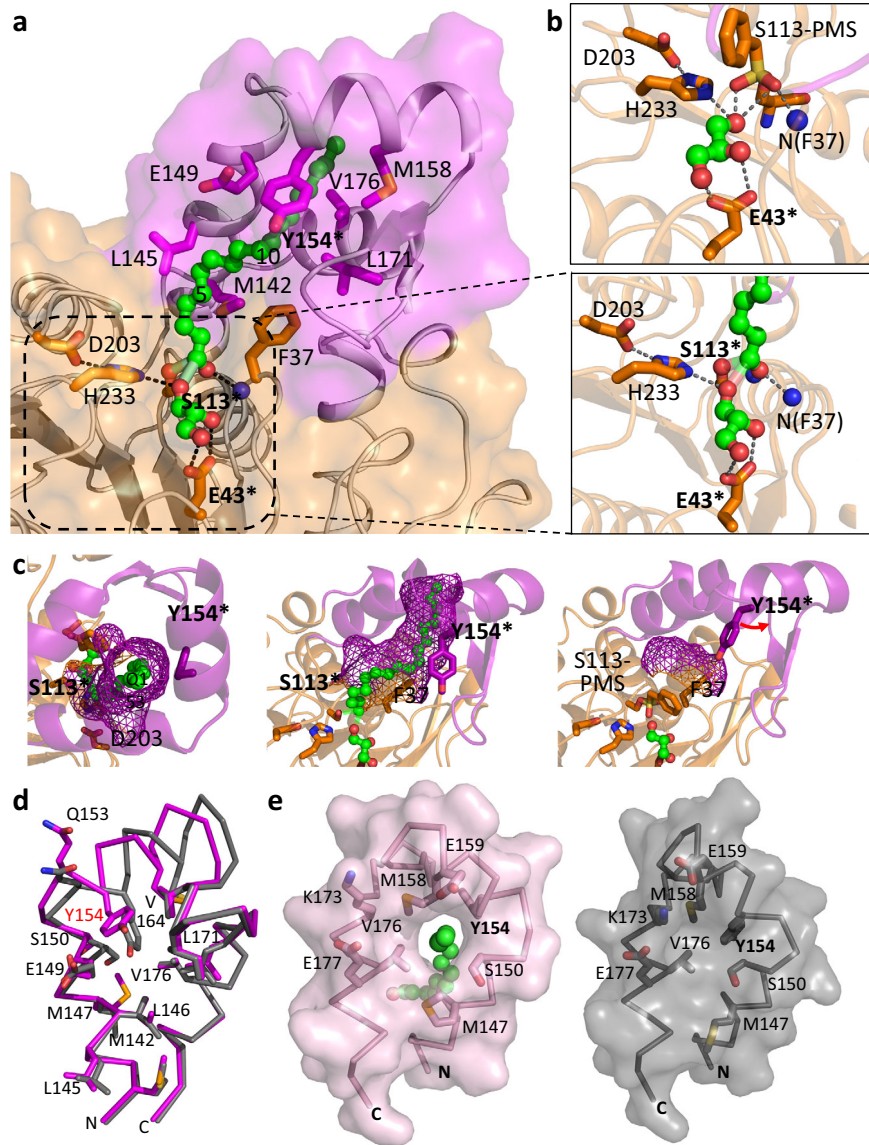

**Fig. 2 | *Tth* MAG lipase active site and lid domain tunnel for long chain MAG esters. a** Surface presentation of the *Tth* MAG lipase long-chain MAG binding site, in part generated by a tunnel across the lid domain. Bound C18 MAG intermediate is shown in green. Atom-specific colors are used for oxygen (red) and nitrogen (blue). Every fifth methylene group of C18 MAG is numbered. Those parts of the C18 MAG that are sequestered by the lid domain tunnel (C11–C18) appear to be darker due to the semi-transparent *Tth* MAG lipase surface. Side chains of key residues contributing to ligand binding and the *Tth* active site are shown and labeled. Residues used for probing function are labeled with an asterisk. The central part of the active site is zoomed in a box to the right, indicating specific polar interactions of the glycerol product with *Tth* MAG lipase Glu43. **b** (Upper inlet), active site of the PMSF-inhibited *Tth* MAG lipase structure for comparison. **c** Mesh surface representation of the C18 MAG lid domain tunnel from the top of the lid domain, indicating that it is open at the most distal end (left panel), side view of the lid tunnel (central panel) similar to the orientation in (**a**). Right panel: for comparison, part of the surface generated by residues Phe37 from the CD and Tyr154 from the lid domain is shown for the distinct conformation of the PMSF-inhibited *Tth* MAG lipase structure, due to a conformational change of Tyr154, which is labeled in red. This change leads to blockage of the lid tunnel found in the *Tth* MAG lipase structure without PMSF. **d** Superposition of *Tth* MAG lipase lid domain structures, without (violet) and with PMSF (gray). **e** Surface presentation of *Tth* MAG lipase lid domain structures, without (violet, left panel) and with PMSF (gray, right panel). Residues involved in the formation of the proximal lid substrate tunnel opening and polar residues forming an outer half-ring wrapping the substrate tunnel exit are labeled. The proximal end of the C18 MAG reaction intermediate bound to the *Tth* MAG lipase structure without PMSF is shown in green. Comparison of the two conformations shows that the lid tunnel is only opened in the *Tth* MAG lipase structure without PMSF, when C18 MAG ester ligand is present. For reasons of optimizing illustration of the lid tunnel the orientation is different from the one in panel (**b**).

spectrometry (LC-MS). A single peak found in the elution diagram corresponded to the glycerol monostearate (2,3-dihydroxypropyloctadecanoate) (Supplementary Fig. 3b), suggesting that under the given sample preparation and crystallization conditions *Tth* MAG lipase products cannot be released.

Based on these data, we modeled a monostearate (C18) acyl ligand into the electron density of the active *Tth* MAG lipase structure (Fig. 2a and Supplementary Fig. 3a). Since we observed significant connecting

density between the terminal C1 carbon of the C18 monoacyl ligand and both the γ-hydroxyl oxygen of Ser113 and the O1 oxygen of the glycerol moiety, unrestrained refinement produced distances between these atoms in the range of 2.2–2.6 Å. The resulting model, which was confirmed by an omit electron density map, supports the presence of a loosely coordinated tetrahedral MAG reaction intermediate[14,33,34], closely matching the geometry of covalently bound tetrahedral reaction mimics in a previously established MAG lipase[12] (Fig. 3 and

**Fig. 3 | *Tth* MAG lipase reaction scheme.** Reaction intermediates are indicated. Color codes: acyl group, pink; glycerol group, blue; reactive water, green; side chain of Glu43, red. The ligand density found C18 MAG ester reaction intermediate found in the *Tth* MAG lipase structure active site (Supplementary Fig. 3) is closest to loosely coordinated tetrahedal transition intermediate 1 (boxed).

Supplementary Fig. 4). We also noticed that a conserved solvent molecule required for ester hydrolysis next to the active site triad residue H233 detected in those structures[12] is missing in our *Tth* MAG reaction ligand complex. As *Tth* MAG lipase is exposed to an acidic water/organic solvent mixture in LC-MS experiments after its re-solubilization from crystals favoring esterification[35], our most plausible explanation is a reverse process from the C18 reaction intermediate observed in the crystal structure to glycerol monostearate substrate detected in LC-MS experiments. All subsequent experiments were designed by interpreting our structural and mass spectrometry-based findings as C18 MAG.

The distal part of the C18 MAG reaction intermediate involving carbon atoms C11 to C18 is held by a hydrophobic tunnel across the *Tth* MAG lipase HBH lid domain, which is open at both ends (Fig. 2a, c, e). Whereas the proximal opening towards the *Tth* MAG lipase active site can be viewed as an extension of the active site area (Fig. 2a), the distal lid tunnel exit is exposed and defined by a ring-like structure of side chains from hydrophobic residues (Fig. 2d, e). In the presence of the C18 MAG reaction intermediate, there are significant alterations in the packing arrangement of several residues within the lid core when comparing to the PMSF-inhibited structure (Fig. 2c–e). The largest spatial deviations are found in the N-terminal part of the lid domain, which includes the first two lid helices α-LA and α-LB (Fig. 2d, Supplementary Fig. 8). In the *Tth* MAG lipase ternary complex structure these two helices are separated by a short loop, whereas in the PMSF-inhibited structure both merge into one long helix (residues 142–161) with a sharp kink at residue Gln153 (Fig. 2d). As a result, in the PMSF-inhibited structure the side chains of Ser150 and Tyr154 next to the kink block the formation of the lid tunnel (Fig. 2c–e). In the presence of the C18 MAG reaction intermediate, the aromatic ring plane of Tyr154 becomes part of the lid tunnel surface. In conclusion, our structural data are strongly suggestive of the hydrophobic lid tunnel to be crucial for positioning long-chain MAG esters to allow efficient turnover by lipase hydrolysis.

## Mechanism for variable length MAG ester turnover

To investigate our structure-based findings suggesting the *Tth* enzyme to function as a MAG lipase, we first measured MAG, DAG, and TAG ester hydrolysis using an established coupled in vitro assay[12]. For C8 MAG, we found a turnover of 16.4+/– 1.3 U mg$^{-1}$ by the wild-type (wt) *Tth* enzyme. In contrast, only residual turnover was observed for the corresponding C8 DAG and TAG esters (Fig. 4b, left panel, Supplementary Table 3), demonstrating that the *Tth* enzyme is indeed highly specific for MAGs. Furthermore, we only found residual activity for long chain C18:1 MAG, presumably due to limited solubility.

We then screened a number of *Tth* MAG lipase mutants for quantitative analysis using C8 MAG as substrate, (Fig. 4b, right panel; Supplementary Table 3). As expected, no activity was found for the S113A active-site mutant, which was subsequently used as negative control. Testing further *Tth* MAG lipase mutants without the lid domain (Δ140–183) and abolishing dimerization (E72R) also resulted in no measurable activity, demonstrating the enzyme to be highly sensitive to structural alterations beyond the catalytic center (Fig. 2a, b). We also probed two further residues, we considered to play roles in MAG turnover. The first one was Glu43, which directly interacts with the O1 and O2 positions of the glycerol moiety (Fig. 2b) and thus most likely contributes to the high level of MAG specificity as opposed to DAG and TAG substrates. While we found about 1/3 of wt activity for the E43A variant, mutation of the same residue to lysine (E43K) led to complete abolishment of MAG activity. This is supported by our structural data (Fig. 2b), suggesting that steric interference with the intricate interaction network between the glycerol moiety of the MAG ester and residues from the *Tth* enzyme may impair its proper positioning and thus substrate turnover. In contrast, removing the tyrosine

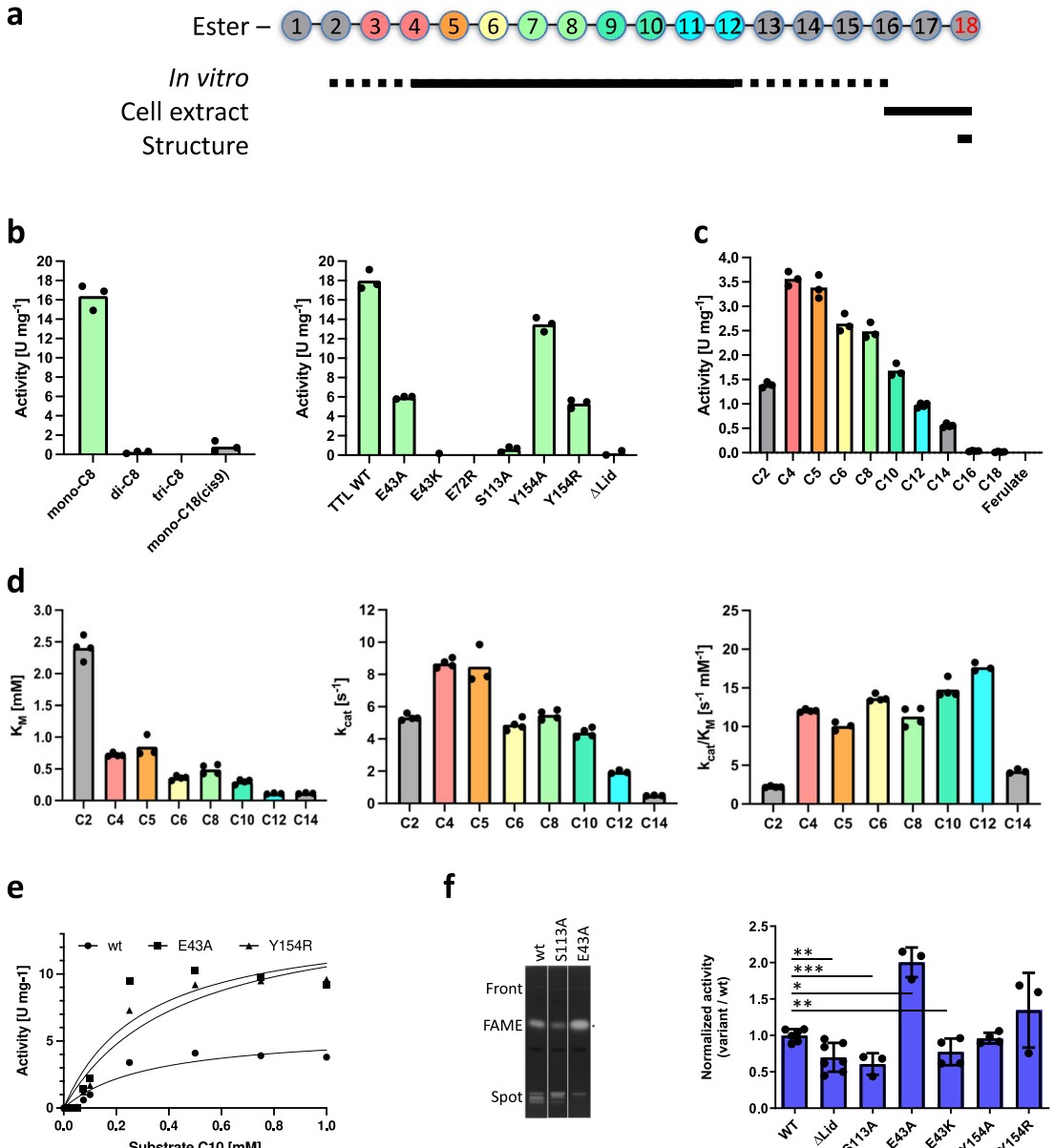

**Fig. 4 | *Tth* MAG lipase activities for MAG esters. a** *Tth* MAG lipase activities for MAG esters of variable length. Coloring scheme of esters characterized in vitro assays (**b**−**d**). MAG esters found in cell extracts (**f**) and by structural data (Fig. 2, Supplementary Fig. 3) are also indicated, demonstrating complementarity of different experimental approaches. **b** Left panel: in vitro turnover of C8 MAG, DAG and TAG esters and C18:1 MAG. **b** Right panel: C8 MAG turnover of different *Tth* enzyme variants. **c** *Tth* MAG lipase activity for several *p*NP acyl esters of C2-C18 FAs and ferulate, indicating a broad range of possible substrates. **d** Steady-state Michaelis-Menten analysis for MAG esters with quantifiable activity profiles. Left panel, apparent substrate binding affinities for C2-C14 MAG esters measured by Michaelis constants $K_M$, indicating increase of binding with increasing MAG ester length; central panel, catalytic rates ($k_{cat}$) for C2-C14 MAG esters, showing an overall profile comparable to that measured for overall substrate turnover (central panel, cf. panel **b**, left); right panel, catalytic efficiencies $k_{cat}/K_M$ for the same substrate spectrum, showing only minor variations for most of the MAG esters used (C4-C12). **e** Left panel: experimental steady-state kinetic data of selected *Tth* variants (wt,

Y154A, E43A) to estimate Michaelis-Menten values for C6 MAG esters. **f** Fatty Acid Methyl Esters (FAMEs) analysis by high-performance thin-layer chromatography (HPTLC) of *E. coli* total lipid extracts (TLE) expressing *Tth* MAG lipase. Left panel, 1, *Tth* MAG lipase (wt); 2, *Tth* MAG lipase (S113A); 3, *Tth* MAG lipase (E43A). Right panel: FAMEs quantification for selected *Tth* MAG lipase mutants, normalized to *Tth* MAG lipase (wt). Data statistics (**b**–**e**): Data are presented as mean values and individual data points, $n = 3$–4 (technical replicates). Data statistics (**f**): Data are presented as means ± SD, and individual data points. Error bars represent the standard deviation from biologically independent replicates ($n = 3$ for variants S113A, E43A, Y154R; $n = 4$ for variants E43K, Y154A; $n = 6$ for WT; and $n = 7$ for ΔLid). Unpaired *t*-test was used to evaluate statistical significance applying *p* values for a two-tailed confidence interval as follows: $p < 0.001$ (***), $p < 0.01$ (**), $p < 0.05$ (*). Definition *p* values: $p = 0.0055$ ΔLid, $p = <0.0001$ S113A, $p = 0.0288$ E43A, $p = 0.0013$ E43K, $p = 0.4985$ Y154A, $p = 0.1285$ Y154R. Source data and uncropped image are provided as a Source Data file.

group of Tyr154 (Y154A) or adding another bulky side chain (Y154R), designed to test possible effect by residues from the *Tth* lid tunnel (Fig. 2c−e), yielded only relative minor activity effects. These findings suggest that there is no significant contribution of the lid to the

turnover of C8 MAG, which is supported by our structural observations that only the distal part of the long-chain C18 MAG is bound by the *Tth* lid tunnel (Fig. 2a). All *Tth* MAG lipase variants were validated by circular dichroism to be properly folded as well as analyzed for thermal

stability and dimeric assembly (Supplementary Tables 1 and 4, Supplementary Figs. 9 and 10).

For quantitative characterization of *Tth* MAG lipase in vitro activity, we subsequently employed a spectrophotometric monoacyl *p*-nitrophenyl acyl ester (*p*NP) assay[28,36]. We used this assay for investigating substrates with variable acyl length ranging from C2 to C18, which is equivalent to the C18 MAG reaction intermediate we found in the *Tth* MAG lipase structure (Fig. 4a), as a convenient approach for direct detection of turnover and thus allowing reliable quantification. We observed a skewed activity profile for C2-C18 esters, with the highest activities found for C4-C8 esters (Fig. 4c, Supplementary Table 3). For C16 and C18 esters, there was only residual activity at a level where quantitative significance estimates were not possible. When directly comparing C8 *p*NP ester hydrolysis with C8 MAG turnover (Fig. 4b, left panel), the activity was about 15%, indicating that the presence of an unnatural *p*NP leaving group replacing the glycerol leaving group in MAG substrates could have a diminishing effect on *Tth* enzyme substrate turnover.

At this point, we suspected that the established *Tth* activity profiles are largely convolutes of diverting length-dependent substrate binding affinities and catalytic turnover. To address this question, we employed steady-state Michaelis-Menten kinetics for quantitatively determining catalytic efficiency $k_{cat}/K_M$, again using the *p*NP ester turnover assay due to its superior statistical reliability (Fig. 4d, e; Supplementary Table 6). We found stronger binding affinities with increasing *p*NP ester length in wt *Tth* MAG lipase, ranging from residual C2 ester binding (>2 mM) to substantially higher binding affinities of around 0.1 mM for the longest measurable *p*NP esters (C12, C14) (Fig. 4d, left panel). As expected, the catalytic rate constants ($k_{cat}$) for C2-C14 *p*NP esters show a similar profile as for overall activity turnover measurements (Fig. 4d, central panel; Supplementary Table 6). As a result, the catalytic efficiencies for C4-C12 *p*NP esters are all in the same range of 12–21 $s^{-1}$ $mM^{-1}$, with insignificant differences of less than a factor of 2 (Fig. 4c, right panel; Supplementary Table 6). The turnover efficiency was considerably less for the shortest substrate used in these experiments (C2) and medium- to long-chain *p*NP esters (> C12), indicative that the catalytic efficiency of medium- to long-chain MAG esters is hampered due to insufficient solubility under the given experimental conditions (Supplementary Table 5).

Finally, we investigated the release of FAs by different *Tth* MAG lipase variants expressed in *E. coli* cells through the analysis of their respective fatty acid methyl esters (FAMEs). When analyzing the FA composition using gas chromatography–mass spectrometry (GC-MS), wt and S113A mutant *Tth* MAG lipase as well as void vector displayed the same composition with mostly hexadecanoic (palmitic) acid (C16:0), cis-Δ11-octadecenoic (cis-vaccenic) acid (C18:1) and smaller contributions of ethylenic (C16:1)/cyclopropanic C17 (9–10 methylene hexadecanoic) and C19 (11-12 methylene octadecanoic) acids (Supplementary Fig. 11). Further, we used high performance thin layer chromatography (Fig. 4f, left panel; Supplementary Fig. 11) to assess the FAMEs released in MAG lipase variants. When comparing the data from wt *Tth* MAG lipase with the S113A negative control variant, wt *Tth* MAG lipase displayed about twofold higher FA amounts, demonstrating the enzyme's ability to turn over the respective acyl esters under cellular conditions (Fig. 4f, right panel; Supplementary Table 7). Interestingly, the *Tth* MAG lipase E43A variant showed a significant two-fold increase of FA turnover with respect to wt *Tth* MAG lipase, while the same *Tth* variant in the in vitro MAG turnover assay showed reduced activity when compared with the wt enzyme. In agreement with our in vitro data as well, expression of the (Δ140–183) *Tth* MAG lipase variant without the lid domain led to only residual level of FA turnover, similar to the S113A control mutant. Interestingly, the *Tth* MAG lipase Y154R variant also showed a tendency for increased FA turnover.

## Tth MAG lipase is a minimal prototype for HBH lid containing lipases

Moving beyond the specific functional properties of *Tth* MAG lipase, we wondered about the enzyme's relations to other previously characterized lipases and esterases. A search against the UniProtKB50 protein sequence data base revealed the top scoring 1000 sequence families to be almost exclusively from bacterial and archaeal phyla (Supplementary Fig. 12a, Supplementary Data 1). Amongst these sequences were only two with known 3D structures: cinnamoyl esterase from *Lactobacillus johnsonii* (UNIPROT code D3YEX6, PDB code 3S2Z, plus additional codes of the same sequence), and feruloyl esterase from *Butyrivibrio proteoclasticus* (UNIPROT code D2YW37, PDB code 2WTM, plus additional codes of the same sequence). Cinnamoyl and feruloyl esters are found as CoA ester intermediates in phenylpropanoids metabolic pathways[37]. Even though these enzymes share a high degree of similarity with *Tth* MAG lipase minor active site sequence differences can explain the inability of *Tth* MAG lipase to hydrolyze phenylpropanoid ester substrates.

The relations of cinnamoyl esterase from *Lactobacillus johnsonii* (UNIPROT code D3YEX6, PDB code 3S2Z, plus additional codes of the same sequence) and feruloyl esterase from *Butyrivibrio proteoclasticus* (UNIPROT code D2YW37, PDB code 2WTM, plus additional codes of the same sequence) to the sequence and structure of *Tth* MAG lipase are at a comparable level, as illustrated by rmsd values of 1.2–1.3 Å and structure-based sequence identities of 30–32% (Fig. 5, Supplementary Fig. 5, Supplementary Table 8). The lid domain in these three structures comprises a virtually identical two-stranded antiparallel β-hairpin (residues 163–165 and 170–172 in the *Tth* MAG lipase), which is connected by a short hairpin motif of five or less residues (Figs. 1c, 5a). This hairpin shields a highly conserved histidine-glycine-phenylalanine (HGF) motif within the catalytic domain loop connecting β-strand 3 and α-helix B (residues 35-37 in the *Tth* MAG lipase structure) and the small α-helix B′ (residues 77–79 in the *Tth* MAG lipase structure) next to the active site (Fig. 5b, Supplementary Fig. 5). Helix B′ comprises a phenylalanine-serine-glutamate/aspartate (FSE/D) motif that is conserved in the sequences of the related esterases from *L. johnsonii, B. proteoclasticus*, and *T. thermohydrosulfuricus* (*Tth*) but not in other related esterase/lipases (Supplementary Fig. 5). Phe37, Phe77 and Phe/Tyr175 from the C-terminal lid domain helix α-LC form a highly conserved aromatic residue cluster in these structures and play a crucial role in shaping the substrate-binding sites of the respective enzymes (Figs. 2a, 6, Supplementary Fig. 5).

Based on these relations it is surprising that *Tth* MAG lipase did not show measurable activity against phenylpropanoid FA ester substrates of the esterases from *L. johnsonii* and *B. proteoclasticus* (Fig. 5a)[38,39]. Inspection of their structures revealed a plausible explanation of the distinct substrate specificities: First, in these esterases, there is an additional second small β-hairpin interaction following the first lid helix α-LA, which is not found in any other HBH esterase/lipase lid domains including the structure of *Tth* MAG lipase (Figs. 5a, 6). This hairpin hovers over the central part of the active site of these two esterases to shield bound substrates and thereby further restricts the size of the substrate binding pocket. In the *Tth* MAG lipase this hairpin structure is replaced by a second lid helix α-LB, which is more distal to the enzyme's active site (Fig. 6a). Secondly, there are two residues in the *Tth* MAG lipase lid structure, Leu145 from the N-terminal lid helix α-LA and Phe175 from the C-terminal lid helix α-LC that are replaced by residues with functional polar groups, aspartate (Asp138, only in the sequence of the *L. johnsonii* esterase sequence) and tyrosine (Tyr169 in the *L. johnsonii* and *B. proteoclasticus* esterase sequences) respectively (Fig. 6). Structures of the esterase from *L. johnsonii* in the presence of several ferulate derivatives (3PFB, 3PFC, 3S2Z) demonstrate the distinct roles of these residues in substrate binding, and removal of the polar side chain of Asp138 strongly impairs substrate turnover[39]. In conclusion, the absence of these functional residue groups in the *Tth*

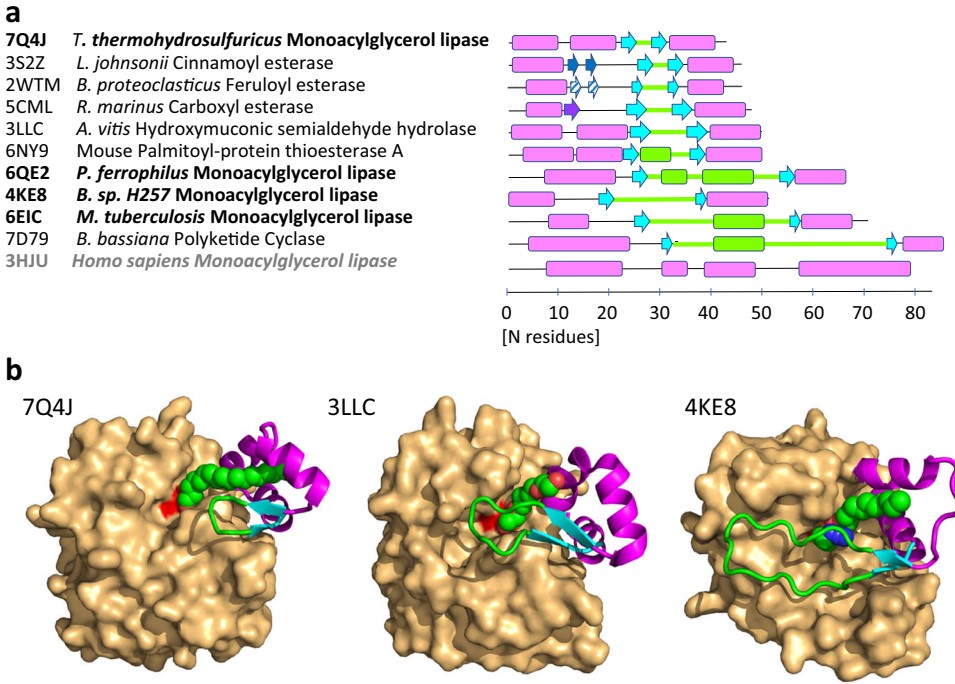

**Fig. 5 | Sequence, structural, and functional relations of esterase/lipases with divergent HBH lid domains. a** Scheme of HBH lid domain sequences related to *Tth* MAG lipase (Fig. 1c) with secondary structural elements indicated proportional to sequence length. The two-stranded HBH lid β-hairpins are in cyan, sequence, and secondary structural elements inserted between strands βL1 and βL2 are in green. The additional two-stranded β-hairpin structure hovering over the active site in the esterase from *L. johnsonii* (PDB code: 3S2Z) is shown in dark blue. The related β-hairpin-like structure in the esterase from *B. proteoclasticus* (PDB code: 2WTM) is shown in the same color with a diagonal stripe pattern. The lid structure of the Osmotically inducible protein C from *R. marinus* (PDB code: 5CML) comprises an additional third β-strand colored in dark violet, which interacts with the first β-strand of the HBH β-hairpin. Human MAG lipase (3HJU), which does not have a HBH lid motif and presents a canonical MAG lipase, is shown for comparison as well. Further details are summarized in Supplementary Table 8. **b** HBH lid structures of

three representative esterase/lipases with their PDB codes indicated (Supplementary Fig. 12b), demonstrating common elements (N- and C-terminal α-helices, two-stranded β-hairpin) and diversity of the inserted sequences in terms of length and structure. The CDs of these structures are shown by surface presentation in light orange, and the HBH domains are illustrated by cartoon presentation. The surface covered by the catalytic triad serine (Ser113 in *Tth* MAG lipase) is shown in red, as a point of reference to locate the active site. Active site ligands are shown in sphere presentation in atom-specific colors (carbon, green; oxygen, red; nitrogen, blue). Only in *Tth* MAG lipase, the C18 ligand crosses the lid domain via a hydrophobic tunnel. The ligands 3LLC (tetraethylene glycol) and 4KE8 (tetradecyl hydrogen (R)-(3-azidopropyl)phosphonate, chain A) are in similar positions next to the active site but they are too short to cross the respective lid domains (Supplementary Fig. 4a).

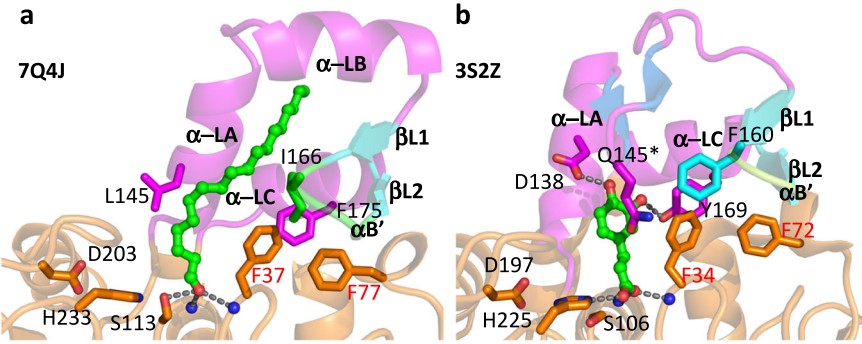

**Fig. 6 | Structural comparison of the ligand bound HBH lid domains and active site area, indicating different substrate specificities. a** *Tth* MAG lipase in the presence of C18 MAG (left); **b** esterase from *L. johnsonii* (PDB code: 3S2Z) in the presence of caffeine (right). Two residues critical for specific hydrogen bond interactions with ferulate in the structure of the esterase from *L. johnsonii* (Asp138, Tyr169) are substituted by hydrophobic residues in *Tth* MAG lipase (equivalent residues Leu145, Phe175). Other key residues with critical roles for substrate binding are also shown. In the structure of the esterase from *L. johnsonii*, an additional

N-terminal lid β-sheet is formed (colored in blue) that hovers over the CD active site. At the end of the short turn connecting the two β-strands, there is a highly exposed residue, which is Gln145 in this structure. Residues that are part of the conserved HGF motif (residues 35-37 in *Tth* MAG lipase, residues 32–34 in the *L. johnsonii* esterase) and less conserved FSE/D motif (residues 77–79 in *Tth* MAG lipase, residues 70–72 in the *L. johnsonii* esterase) are labeled in red (Supplementary Fig. 2).

MAG lipase well explains the lack of any measurable activity towards hydroxycinnamic esters.

Next, we searched for more distantly related esterase/lipase sequences by exploiting available structural data. The structure-based identities of all 16 selected target sequences (except those from *L. johnsonii* and *B. proteoclasticus* with closer relations) are in the range of 15-23 % and the root mean square deviation values of the superimposed structures are between 1.9 and 2.2 Å, suggesting their relations to be distant with uncertain phylogenetic relations (Supplementary Fig. 12b, Supplementary Table 8). Invariant residue positions are confined to the active site including the catalytic triad and four glycine positions (Gly71, Gly111, Gly115, Gly116), which also contribute to the active site oxyanion hole, another hallmark of α/β hydrolases (Supplementary Fig. 5)[24,40].

Remarkably, we found an accumulation of esterase/lipase sequences with non-canonical HBH lid domains (Fig. 5, Supplementary Table 8, Supplementary Fig. 5). This motif is not present in most previously characterized MAG lipases that comprise lid domains with one or more helices without a β-sheet inserted[22], we define here as canonical. These include human MAG lipase, which has been used as model in various studies[15,41] (Fig. 5a). Comparison of these HBH lid structures showed that the terminal helices and adjacent loops function as flexible hinges, even in structures of the same sequence with different ligands (Fig. 5b)[22,24]. On the other hand, the hydrogen bond pattern found in the conserved lid β-hairpin provides an additional element of local fold rigidity, which is not found in other more generic lid domains that are formed from α-helices only. While the small antiparallel β-hairpin motif is conserved in HBH lid-containing esterases/lipases, there is substantial divergence with respect to the length of inserted sequences (Fig. 5). Different from the short hairpins found in the *Tth* MAG lipase and the most closely related esterases from *L. johnsonii* and *B. proteoclasticus*, in other more remotely related sequences there are various levels of extensions, allowing to cover extensive surface areas of the CD active site. The most extreme case is found in the hydrolase-1 domain-containing protein from *B. bassiana*, which contains an insert of 43 residues with a 12-residue helix and an extensive loop structure, allowing to shield the complete active site of this enzyme (PDB code 7D79) (Fig. 5a, Supplementary Table 9).

We finally investigated to what extent esterase/lipases with a HBH lid domain may resemble properties we observed in the structure of *Tth* MAG lipase. Remarkably, none of the structures with a short HBH lid domain (Fig. 5a), except for *Tth* MAG lipase, reveals any tunnel-like structure across the lid domain. This is in accordance with data indicating most efficient substrate turnover for esters of small organic compounds with additional polar groups, as well demonstrated for hydroxycinnamic acid family esterases[38,39]. In contrast, HBH lid tunnels are observed in at least two structures with long HBH lid domains the *Paleococcus ferrophilus* MAG lipase (6QE2) and the *Bacillus* sp. H257 lipase (4KE8/4KE9), which both have been annotated as long-chain MAG lipases (Supplementary Table 8). In one of these structures of the MAG lipase from *Bacillus* sp. (4KE9), the distal end of a 16-carbon MAG analogue binds through a short tunnel of its HBH lid domain[12], with some resemblance of the HBH lid path found in the structure of *Tth* MAG lipase. As key residues of the *Bacillus* sp. enzyme involved in regulation of MAG lipase turnover (Ile145, Glu156) are positioned on segments of the long HBH lid loop insert, which is missing in *Tth* MAG lipase (Fig. 5b), the precise mechanism of MAG lipase turnover regulation appears to be unrelated. A more detailed structural comparison is not possible as there is only very little conservation of residues contributing to long-chain MAG binding.

Taking the data together, HBH lid domains seem to play a crucial role in esterase/lipase function. Supported by our *Tth* MAG lipase structural and functional data, they promote binding of especially long chain MAG substrates. MAG lipase binding and turnover is promoted by a toolbox of specific lid domain features: largely varying levels of extensive substrate-shielding areas as demonstrated by vastly diverting HBH β-hairpin sequence inserts, and the formation of lid domain tunnels to hold especially the distal part of long-chain MAG esters. Ultimately, our structural data on *Tth* MAG lipase demonstrate that the presence of a minimal HBH lid domain is sufficient to allow turnover of long-chain MAG esters. Thus, the structure of *Tth* MAG lipase presents a minimal prototype for a growing number of esterases and lipases with HBH lid domains of different length. Our data illustrate that a profound understanding of MAG lipase catalysis goes well beyond classical enzymology with well-defined active site/substrate relations. Knowledge on enzyme-specific contributions by lid structure and dynamics next and topping the active site are essential for our understanding of substrate specificity and possible future applications on lipase design with further improved turnover properties. Our structure data also reveal the importance on the way how the glycerol backbone of lipase acyl substrates is bound by specific active site residues, ultimately defining its activity profile as MAG lipase, which is not permissive to act as a DAG or TAG lipase. HBH lid lipases with the ability for tunnel formation, as defined in this contribution, may offer distinct opportunities for additional evolvement to recognize promiscuous lipidic substrates of variable length, which may even be extracted directly from lipid membranes[21,41].

In conclusion, this study presents a rare case where a high-resolution enzyme structure in the presence of an endogenous long-chain lipid reaction intermediate led to its functional annotation as a MAG lipase. We attribute the opportunity of capturing such complex to the extremophilic properties of its natural host *Tth*, which are sufficiently different from the growth conditions of the target enzyme's heterologous expression host (*E. coli*). Due to the high level of lid diversity and dynamics, homology modeling tools successfully advocated for various other discoveries of enzymatic function[1,2] would have failed, as the closest related enzymes with known structures to *Tth* MAG lipase have specific phenylpropanoid FA esterase activities. Ultimately, this study highlights the strength of high-resolution structural biology to identify and characterize unexplored enzymatic functions under physiological conditions that allowed effective inhibition of substrate turnover. It may inspire future investigations of enzyme function by systematically searching for suitable conditions, under which endogenous substrates could be directly captured.

## Methods

### Protein expression and purification

For X-ray crystallography studies, wt *Tth* MAG lipase was expressed and purified according to a previously established protocol[28]. In this contribution, the protein was expressed using a pETBlue-1 (Novagen) vector in *E. coli* (DE3) strain. The purification was performed by liquid chromatography using a Sepharose column followed by Size Exclusion Chromatography (SEC) step. For SEC profile analysis, a Superdex 75 increase 3.2/300 column (Cytiva) was used. The protein was eluted in a 50 mM Tris-HCl pH 8.0 buffer containing 170 mM NaCl and the pure fractions were pooled together and concentrated up to 20 mg ml$^{-1}$. For production of the seleno-L-methionine (SeMet)-incorporated *Tth* MAG lipase, the corresponding expression vector was used to transform the methionine auxotroph *E. coli* strain B834(DE3). A preculture grown in LB broth medium was used for the expression of the protein in 2 L M9 minimal medium including trace elements. After a 3-h starvation step a L-aminoacid mixture was added (40 mg ml$^{-1}$ each except methionine) and SeMet (60 mg ml$^{-1}$). The culture was induced with 1 mM isopropyl-β-D-thiogalactopyranoside (IPTG) when entering the logarithmic growth phase and was further grown overnight. The SeMet protein was purified using the same protocol as the native protein.

To generate different protein variants for in vitro assays we subcloned the gene encoding *Tth* MAG lipase into a pETM-14 vector (EMBL). For all single residue *Tth* variants used in this study, we performed PCR-based mutagenesis on wt *Tth* MAG lipase. For the *Tth*

MAG lipase (Δ140–183) construct two primers were designed, the first one in forward orientation for the C-terminal segment of the catalytic domain expressing residues 183–259 and the second one in reverse orientation for the N-terminal segment of the catalytic domain expressing residues 1–140 (Supplementary Table 9). Both primers generated a unique KpnI restriction site. PCR amplification followed by KpnI restriction and ligation yielded the *Tth* MAG lipase (Δ140–183) construct.

The genes encoding wt *Tth*, E43A, E43K, E72R, Y154A, and Y154R were cloned into the pETM-14 expression vector, and for expression *E. coli* BL21-CodonPlus(DE3)-RIL cells (Agilent Technologies) were used. Cells were grown at 37 °C in lysogeny broth (LB-Lennox) supplemented with kanamycin and chloramphenicol. Cells were induced by the addition of 0.5 mM IPTG at an $OD_{600}$ of 0.6–0.8 and harvested after continuous shaking overnight at 18 °C. For purification, the cells were resuspended in lysis buffer (25 mM Tris-HCl (pH 8.0), 300 mM NaCl, 20 mM imidazole, and DNaseI), sonicated, and centrifuged at $40,000 \times g$ for 30 minutes at 4 °C. The supernatant was filtered and loaded onto Ni-NTA beads, which had been equilibrated in wash buffer (25 mM Tris-HCl (pH 8.0), 300 mM NaCl, 20 mM imidazole). After loading, beads were washed with wash buffer and eluted in elution buffer (25 mM Tris-HCl (pH 8.0), 300 mM NaCl, 500 mM imidazole). Each *Tth* variant was concentrated using an Amicon® Ultra concentrator with a 10 kDa cutoff (membrane-regenerated cellulose) and further purified by size exclusion chromatography, using a Superdex 75 10/300 column (GE Healthcare), equilibrated in 25 mM Tris-HCl (pH 8.0), 50 mM NaCl.

## Tth MAG lipase reductive methylation

*Tth* MAG lipase, purified protein was concentrated to 5 mg ml⁻¹ and for each 1 ml of enzyme solution, 20 µl 1 M dimethylaminoborane (DMAB) was added followed by 40 µl 1 M formaldehyde, as previously described[42]. DMAB and formaldehyde were added once more after 2 h of incubation. After another 2 h, another 10 µl 1 M DMAB was added and incubated for 18 h. During the entire incubation time the sample was kept in the dark and at 4 °C. To terminate the reaction, DMAB and formaldehyde were removed via a three-step centrifugation buffer exchange to a 50 mM Tris-HCl pH 8.0 and 100 mM NaCl buffer using a filter with a 10 kDa molecular-weight cutoff. Methylated and non-methylated *Tth* MAG lipase were compared by SDS-PAGE (Supplementary Fig. 13). The level of methylation of *Tth* MAG lipase was quantified by quadrupole time of flight (Q-TOF) mass spectrometry (Supplementary Table 11).

For these experiments, protein samples were acidified using 1% trifluoroacetic acid prior to injection into the Acquity UPLC System (Waters Corporation). Approximately 2 µg of each sample were loaded onto a protein separation column (Acquity UPLC Protein BEH C4 column, 2.1 mm × 150 mm, 1.7 µm). The outlet of the analytical column was coupled directly to a Q-TOF Premier mass spectrometer (Waters/Micromass) using the standard Electrospray Ionisation source in positive ion mode. Solvent A was water, 0.1% formic acid, and solvent B was acetonitrile, 0.1% formic acid. For Q-TOF experiments, a spray voltage of 3.5 kV was applied with a cone voltage of 35 V and extraction cone at 5 V. The desolvation temperature was set at 350 °C, with source temperature 120 °C. Desolvation gas was nitrogen at a flow rate of 500 L/min. The collision energy was set at 5 eV with argon in the collision cell at a pressure of $4.5 \times 10^{-5}$ mbar. Data were acquired in continuum mode over the mass range 500–3500 m/z with a scan time of 0.5 s and an interscan delay of 0.1 s. Data were externally calibrated against an intact protein reference standard (Thermo Fisher), acquired immediately prior to sample data acquisition. Spectra from the chromatogram protein peak were summed and intact mass was calculated using the MaxEnt1 maximum entropy algorithm (Waters/Micromass) to give the zero charge deconvoluted molecular weight.

Catalytic activity against C8 *p*-nitrophenyl MAG ester using the assay as described in section "In vitro assay to determine lipolytic enzyme activity" was close to non-methylated *Tth* MAG lipase (2, 3 U mg⁻¹, cf. Fig. 4b, left panel, and Supplementary Table 3).

## Tth MAG lipase crystallization

Crystallization experiments using commercial screens for the sitting drop vapor diffusion technique were performed at the EMBL-Hamburg crystallization facility at 20 °C[43]. Protein solution at 20 mg ml⁻¹ was mixed with the precipitant solution to a final volume of 0.4 nl (0.2 + 0.2 nl). Single rod-shaped crystals of about 100 µm in length grew from precipitant solution containing 5% PEG-6000 in 0.1 M Tris-HCl (pH 8.0) buffer.

We also used a methylated version of the protein at 20 mg ml⁻¹ for *Tth* MAG lipase crystallization. Well-diffracting crystals with a maximum size of 50 × 50 × 100 µm appeared after 7–10 days using precipitant solution of 2.4 M ammonium sulfate and 0.1 M Tris (pH 7.5) buffer. For the PMS-bound structure, the protein was incubated for 2 h with 10 mM PMSF followed by the same crystallization protocol. Identical crystallization conditions were used for methylated SeMet-substituted protein.

## X-ray data collection, structure solution, and validation

Prior to X-data collection, the methylated *Tth* MAG lipase crystals were immersed in mother liquor containing 30% (v/v) glycerol and flash cooled in liquid nitrogen. All data sets were collected at 100 K. An anomalous dispersion experiment of a peak ($\lambda = 0.97873$ Å) and an inflection ($\lambda = 0.97903$ Å) data-set using a SeMet-containing methylated *Tth* MAG lipase crystal was carried out on the tunable beamline BM14 (EMBL/ESRF, Grenoble, France). X-ray data of PMSF-inhibited *Tth* MAG lipase were also collected on the same beamline ($\lambda = 0.95372$ Å). The SeMet crystal belongs to the space group P4₃2₁2 while the PMS-bound to the space group P4₃22, both estimated to contain a protein dimer per asymmetric unit.

All data sets were integrated, scaled, and merged using the HKL suite[44]. Selenium atoms were located using the SHELXD program and initial density modification was performed by SHELXE[45]. The correct heavy atom substructure was then used together with the anomalous data from the HKL suite as input for the PHENIX AutoSol wizard, which recalculated the experimental phases and carried out density modification[46]. The density modified phases where then merged with the high resolution data-set from the same SeMet crystal using the program CAD and were then used as an input for model building by ARP/wARP[47,48]. The produced model was further improved by manual corrections and modifications in COOT[49]. ARP/wARP was used again for adding ordered solvent molecules. Additional ligands were added manually after inspecting the electron density maps in COOT. Molrep was used to determine the remaining structures (PMS-bound and native)[50]. Refinement using molecular dynamics and TLS was performed by PHENIX[51]. The data collection and refinement statistics are reported in Supplementary Table 2.

Native *Tth* MAG lipase crystals were immersed in crystallization solution containing in addition 15% MPD as cryoprotectant and then immediately mounted at the beamline's goniometer cryo-cooled at 100 K. X-ray data to 2.42 Å resolution were collected at a wavelength $\lambda = 0.9763$ Å at the EMBL/DESY beamline BW7A at the DORIS III synchrotron storage ring integrated and scaled as above. We determined initial phases by molecular replacement using the Se-Met substituted and lysine methylated *Tth* MAG lipase coordinates (7Q4J). Structure refinement was carried out as described above with the addition of non-crystallographic symmetry restraints.

The quality of all models was assessed by Ramachandran plots with 97.5%, 97.2%, 97.6% of the residues in the most favored positions for the active Se-Met, PMSF-inhibited and native structures while no residues were present in disallowed positions (Supplementary Fig. 6).

A structure-based sequence alignment was computed by PDBeFold[52]. Figures of the 3D structural model were prepared using PYMOL v.2.5.

AlphaFold2 (ColabFold version1.5.2) was used for modeling the *Tth* MAG lipase structure from the sequence, by forcing a dimeric arrangement. The resulting models superimposed onto the *Tth* MAG lipase crystal structures with a root mean square deviation of about 0.45 Å.

## Analysis of crystal content

About 30 crystals of Se-Met *Tth* MAG lipase were carefully washed with water and dissolved in 200 µl 1 M NaCl. A chloroform/methanol mixture of 250/500 µl was then added to the solution. After 10 min of intensive vortexing, 250 µl chloroform were added and the solution mixed for 1 min. Then, 250 µl water were added and the solution mixed again for 1 min. The mixture was centrifuged for 10 minutes at maximum speed and the upper water phase was carefully discarded. The organic phase was evaporated under vacuum centrifugation at room temperature and resuspended in 20 µl hexafluoroisopropanol. LC-MS measurements injecting 20 µl of the sample in a C18 column and applying a 50–95% acetonitrile gradient in water, revealed a single peak eluted from the column, which in the MS spectrum gave a peak at 359 in positive ionization mode (358+1).

## Monoacylglycerol (MAG) ester assay

Enzyme activity towards MAG esters was measured as follows[53]. The protein concentration of the purified enzyme variants was adjusted with a sample buffer comprising 50 mM Tris (pH 8.0) and 100 mM NaCl to a suitable concentration range of 0.05–0.005 mg ml⁻¹ to perform measurements in the linear range of the assay. 1-Octanoyl-*rac*-glycerol and 1-oleoyl-*rac*-glycerol were used as short- and long-chain MAG substrates, respectively. 1,2-Dioctanoyl-*sn*-glycerol and tricaprylin were used as DAG and TAG substrates. All substrates were purchased from Sigma-Aldrich if not stated otherwise. Prior to use, the substrate stocks were diluted with assay buffer comprising 50 mM Tris/HCl (pH 8.0) to a final concentration of 1 mM and supplemented with defatted bovine serum albumin at a molar ratio of 1:1. All values were determined in triplicates and corrected by blank subtraction. The activity assay was performed at 40 °C for 10 minutes followed by the addition of 100 µl chloroform, centrifugation (10,000 × g for 10 min) and determination of the released glycerol in 10 µl of the upper phase with a commercial Glycerol Assay Kit according to the manufacturer's instructions (Sigma-Aldrich).

## Mono p-nitrophenyl acyl ester (pNP) assay

Cleavage of *p*-nitrophenyl acyl (*p*NP) esters was determined as in previous studies[28]. The protein concentration of the purified enzyme variants was adjusted with sample buffer (25 mM Tris, pH 8.0 and 50 mM NaCl) to a suitable concentration ranging from 0.02 to 0.05 mg ml⁻¹ to perform measurements in the linear range of the assay. The substrates were stored as 10 mM stock solutions in 100 % ethanol protected from light at −20 °C. Prior to use, the substrate stocks were diluted with assay buffer (50 mM Tris-HCl, pH 8.0) additionally containing 1 mg ml⁻¹ gum arabicum (Sigma Aldrich). All values were determined in quadruplicates and corrected for autohydrolysis using the experimentally determined pH-dependent extinction coefficient of *p*-nitrophenol in microtiter plates ($\varepsilon$ = 9.3024 mL µmol⁻¹). The activity assay was performed at 40 °C in a semi-automated high-throughput setup in micro-scale using clear, flat-bottom 96-well plates (Corning). Applying a Freedom EVO® screening robot equipped with a Safire2 plate reader (Tecan Group Ltd.), *p*NP substrates were used at a final concentration of 0.9 mM, and continuous measurements were performed at $\lambda$ = 410 nm for 10 min in a total assay volume of 200 µl per well. Blanks with sample buffer, but without enzyme, were simultaneously measured for each substrate.

*Tth* MAG lipase Michaelis-Menten kinetics were performed by quadruple measurements of enzyme activity with *p*NP esters at different substrate concentrations (0.005, 0.0075, 0.01, 0.025, 0.05, 0.075, 0.1, 0.25, 0.5, 0.75, and 1 mM). The respective activity values were fitted to the Michaelis-Menten equation applying a non-linear regression tool in Prism version 9.1.2. Obtained $K_M$ and $v_{max}$ values of each Michaelis-Menten fit were then used to calculate averages and standard deviations of these basic kinetic parameters. Finally, $k_{cat}$ and $K_M$ yielded the catalytic efficiency (Supplementary Table 6).

## Cellular Tth MAG lipase activity by cell extract analysis

Wt *Tth* and Y154A were cloned into the pETBlue-1 vector, transformed into *E. coli* Tuner™ (DE3) pLacI cells and grown at 37 °C in lysogeny broth (LB-Lennox) supplemented with ampicillin and chloramphenicol. Cells were induced by the addition of 1 mM IPTG at an $OD_{600}$ of 0.6–0.8 and harvested after continuous shaking at 37 °C for 1 h. Cell pellets were washed with PBS prior to lipid extraction.

For lipid isolation and analysis, total lipid extracts (TLE) were prepared from wet *E. coli* cell pellets by three successive extractions using distinct chloroform/methanol mixtures (1:2, 1:1, and 2:1 v/v). The fractions were pooled, washed with water, and dried under air stream. For chromatography analyses, FAs were methylated using TMS-diazomethane (Sigma), purified on a florisil batch column and analyzed by thin layer chromatography (TLC) (Merck silica Gel 60) with dichloromethane as developing solvent.

For lipid semi-quantitative analysis, equivalent masses of total lipid fraction from each strain were spotted onto a high-performance TLC (HPTLC) silica gel 60 plate (Merck) with a Camag ATS4 apparatus. The plates were developed in dichloromethane using a Camag ADC2 device and stained by immersion using a Camag CID3 apparatus with 10% primuline in acetone/water (84:16). Lipids were revealed and quantified by determining their relative percentage using the Chemidoc Imaging System and ImageLab software (Biorad).

To determine FA composition by GC-MS, the FAMEs were purified by preparative thin-layer chromatography (TLC) using a silica gel coated plate G60 (Merck) with dichloromethane as developing solvent. Visualization was performed by spraying the plate with rhodamine B, and FAMEs were scraped off the plate and extracted twice with diethyl ether. The purified FAMEs were then analyzed by GC-MS (Thermo Scientific gas chromatograph, fitted with a 15 m × 0.25 mm TG1MS fused silica capillary column and connected to an ISQ single quadrupole mass spectrometer), and identified according to their retention time and their fragmentation pattern[54].

## Nano differential scanning fluorimetry

Temperature-induced aggregation curves were measured using Prometheus NT.48 (NanoTemper Technologies GmbH) equipped with a scattering module. Proteins were prepared in 25 mM Tris-HCl (pH 8.0), 50 mM NaCl and measured in three technical replicates at a concentration of 1.5 mg ml⁻¹. Excitation power was pre-adjusted to get fluorescence readings above 2000 RFU for F330 and F350, and samples were heated from 15 °C to 95 °C with a slope of 1 °C per min. The processed data were exported from the manufacturer's software (PR.ThermControl) and analyzed with MoltenProt to obtain aggregation temperatures[55].

## Circular dichroism

Far-UV CD spectrums for the *Tth* samples were recorded on a Chirascan Circular Dichroism Spectrometer upgraded with an Active Nitrogen Management System (Applied Photophysics). Measurements were performed at a controlled temperature of 20 °C in a 1 mm QS cuvette, between 280–190 nm using a bandwidth of 1 nm. The *Tth* samples were measured at a concentration of 0.25 mg ml⁻¹ in 20 mM Tris-HCl (pH 8.0) in three technical replicates. Signal from buffer was

subtracted for all measured samples. The estimation of the secondary structure contents was calculated using CONTIN[56].

## Statistics and reproducibility

All enzymatic experiments were carried out at least in triplicate as follows: MAG ester assay: $n = 3$ or 4; $p$NP assay: $n = 3$ or 4; cellular $Tth$ MAG lipase activity by cell extract analysis: $n = 3$, 4, 6 or 7; nano differential scanning fluorimetry: $n = 3$; circular dichroism: $n = 3$. Statistical significance of the cell extract analysis data (Fig. 4f) was evaluated using two-tailed Student's $t$-test and $p$ values of less than 0.05 were deemed significant. Statistical analysis was performed using GraphPad Prism 5.04.

## Reporting summary

Further information on research design is available in the Nature Portfolio Reporting Summary linked to this article.

## Data availability

The coordinates of the X-ray structures determined in this contribution have been deposited in the Protein Data Bank under accession codes 7Q4J, 7Q4H, and 8B9S. All other data generated in this study are provided in the Supplementary Information and a Source Data file. The coordinates of the following X-ray structures used for analyzing the results in this work are also available from the Protein Data Bank with accession codes: 3S2Z, 2WTM, 7D79, 3HJU, 3LLC, 4KE8, 4KE9, 3PFB, 3PFC, 5CML, and 6QE2). Source data are provided with this paper.

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

## Acknowledgements
We acknowledge the technical support by the Sample Preparation and Characterization facility of the EMBL Hamburg Unit, the EMBL Proteomics Core Facility and the Proteomics Core Unit of the University of Münster, Germany. We also acknowledge the contributions of Marina Royter during the early phase of the project, as well as the technical contributions of Anke Peters and Barbara Klippel, all from the Hamburg University of Technology, as well as Lucie Spina and Marie-Antoinette Laneelle from the IPBS, University of Toulouse, CNRS, UPS, for contribution in methodology. We are grateful to Dr Martin Walsh at the European Synchrotron Radiation Facility (ESRF), Grenoble, France, for providing assistance in using the beamline BM14 for our X-ray data collection experiments. We thank Sihyun Sung, Grzegorz Chojnowski and Barbara Ramsak from the EMBL Hamburg Unit for technical and scientific support, and Chris Meier from the University of Hamburg for advice during the manuscript revision process.

## Author contributions
N.P., A.K., H.M. and M.W. designed the project. N.P., A.K., N.T., S.D.C. and C.L. performed the experiments. N.P., A.K., S.D.C., S.A.M., H.M. and M.W. analyzed the data. N.P., A.K., S.A.M., H.M. and M.W. wrote the manuscript. M.D., A.G. and M.W. supported the project.

## Funding

## Competing interests
The authors declare no competing interests.
