## [Peer Review File · Nature Communications]

Discovery of a non-canonical prototype long-chain monoacylglycerol lipase through a structure-based endogenous reaction intermediate complexREVIEWER COMMENTS

Reviewer #1 (Remarks to the Author):

This manuscript has described a high resolution structure of monoacylglycerol lipase and the molecular basis for long-chain length substrate recognition have been highlighted. Some questions below are necessary to answer before considering published.

1、 In the “Extremophiles 13, 769-83 (2009).”, the lipase from *Thermoaerobacter thermohydrosulfuricus* show substrate preference toward triacylglycerols with fatty acids in different chain lengths. Is the same enzyme described in this manuscript? The enzyme in this study is a monoacylglycerol lipase not a TAG-hydrolyzing lipase, but the substrate characterization results have not been seen.

2、 Reference 20(*Journal of Biological Chemistry* 288.43 (2013): 31093-31104.) has described the crystal structure-substrate complex of a bacterial monoacylglycerol lipase, and also mentioned some key sites were importance for substrate binding. Please make some comments compared to your enzyme.

3、 Why monoacylglycerol lipase were selected for studying in this work? Is it important for industrial application or life science?

4、 The surface activation phenomenon are common for triacylglycerol lipase. Any evidences has been found in this study for the open and close conformation of lid domain?

5、 Glycerol products have been found this lipase structure. Does it has the relation with the glycerol releasing hole of monoacylglycerol lipase (*Journal of Chemical Information and Modeling*, 2021, 62(9): 2248-2256).

6、 In this work, HBH domain in lid of Tth MGL have found to involve in long-chain length substrate recognition. But, lid domain in lipase have been confirmed for substrate recognition by intense studies. Although it is a nice work with fancy structural results, the finding here it is not so surprising.

Reviewer #2 (Remarks to the Author):

The Authors present nicely done high resolution crystal structures and provide a comprehensive analysis of the structures. Thorough evaluation of those interpretations was done providing enzymatic activity and enzyme kinetics data for mutants of residues deemed important. The Authors report an endogenous reaction intermediate which would be a significant observation. However there are some points unclear

especially surrounding the intermediate that was found.

General comments

The Authors make the point that monoacylglycerol lipases are important for a number of industrial processes. However it would be nice to have more comment on how this particular enzyme is relevant in this sense. There are numerous enzymes of this class known that show high temperature stability with LipS even showing it's highest activity at 70C (doi:10.1371/journal.pone.0047665).

Given the impact factor of the journal it would be good to have more emphasis on how this new found structure and enzymatic activity is relevant. It would be good to have a stronger point on how this enzyme improves on the already known enzymes or how the findings stand out over previous studies. Alternatively a point could be made about the relevance of the enzyme for the survival of the host organism.

The Authors report a reaction intermediate, it is unclear what exactly is meant by this.

Do the authors claim to see the short lived tetrahedral intermediate formed during the enzymatic reaction?

In this case it would be good to have comment on why this intermediate is supposedly stable in this particular case and why the reaction would not go to its end. All that would be needed is to finish the reaction is water which should be abundantly available in a protein crystal available.

It should also be made clearer what reaction intermediate the authors are proposing, this could be marked in Figure 4. If this indeed represents an endogenous intercommunication this is very significant, however there is not much evidence provided for this and liquid chromatography mass spectroscopy seems to indicate the presence of uncleaved substrate.

Specific comments

Page 2 The Authors make the point that this structure would not have been accessible normally due to low solubility of the ligand. However there are structures published with high chain length substrates (PDB 4KE9, 4KE8) indicating that the substrates can be made soluble enough to deliver them to the protein.

Page 3 The Authors make the point that the limited solubility of the substrates make previously reported substrate profiles unreliable. However it is not mentioned how this is not a problem in the study presented here. The solubility of the p-nitro-phenol substrate used by the authors will also decrease with increasing carbon chain length, this is even described in the supplementary section. It would be good to have an explanation why this is considered a problem in previous studies but not in this one.

Page 3 The experimental details described seem a bit slim. It is usual to mention crystallization conditions and at least the final purification buffers. It doesn't seem likely to me the work presented could be reproduced with the experimental details given.

The Authors also make a point of seeing glycerol in their binding pocket even in the presence of PMSF (Page 5). Glycerol is a very common additive during protein purification and crystallization and one of the most popular cryo-protectant during freezing of protein crystals. Without more experimental details it is

difficult to judge whether this glycerol could come from the host organism or the sample preparation. More details on how the enzymatic assays were done would also be good to better interpret the results.

On Page 5 The authors state, they find a single peak for glycerol-monostearate doing mass spectrometry, however in the same sentence they interpret this in a manner that product can not be released. This seems somewhat contradictory a glycerol-monostearate should be a substrate to a monoacylglycerol lipases not product. The Authors also report a reaction intermediate in this manuscript, it seems their mass spectroscopy findings might contradict this. If the fatty acid is in one of the intermediate states it should be contently linked to the enzyme (Figure 4), does that not contradict the mass pectrometry findings?

On Page 7 The Authors mention an unexpected gain of function for a Glu43 variant. This unexpected behavior might be a result of the p-nitrophenol leaving group being handled differently than a glycerol leaving group in the crystal structure. It would be interesting to see the behavior of this variant in an assay using a glycerol ester as substrate.

Figure 2 The Authors claim to have found a reaction intermediate, however in their structure it seems there is a covalent bond missing between the glycerol and the fatty acid chain.

It should be clarified why this bond is missing.

If this is supposed to either be the substrate or the intermediate, this bond should be considered during ligand placement and refinement since it might put a restrain on the geometry of the substrate that is missing without the bond.

Additionally it would be good to have more information about the geometry, depending on what intermediate it is there should be specific angles around the alpha carbon of the fatty acid moiety.

One concern I have is that using this structure as evidence for the intermediate seems insufficient, there needs to be experimental proof that there is a bond between the substrate and the enzyme, especially since a peak for the substrate in mass spectrometry already suggests that this might just be bound substrate that is not contently linked.

Reviewer #3 (Remarks to the Author):

This interesting study presents the structure of an enzyme from the thermophilic anaerobic bacterium *Thermoanaerobacter thermohydrosulfuricus* (Tth) that has first been described in 2009 as LipTth and according to the amino acid sequence and substrate activity against “a broad range of substrates, including triacylglycerols, monoacylglycerols, esters of secondary alcohols, and p-nitrophenyl esters” has been assigned as a lipase. Medium chain Length (C8-C14) had been reported as the preferred chain lengths for p-nitrophenyl ester substrates. LipTth had also been reported to be highly S- stereoselective towards esters of secondary alcohols with high robustness with respect to temperature and different solvents. These features render the lipase potentially interesting for biotechnological studies and the aim

of the present study is to understand enzymes activity, structure and substrate promiscuity using X-ray crystallography combined with biochemical and biophysical studies of wt protein and mutations thereof. In this study, the protein is termed Tth-MGL. Crystal structures were determined in presence of PMSF, and in the absence of PMSF a C18 monoacyl ester intermediate was observed in the active site tunnel, showing a large substrate spectrum. Also of interest is, that crystals were only obtained after methylation and the structures were solved with Se-Met Tth MGL.

Major concerns:

Overall: The data in the manuscript are really interesting and detailed for the MGL community. However the manuscript is difficult to read in some parts and might benefit from a restructuring e.g. aid the reader in the orientation of the structures (1D, E and F all have different orientations); the colors in 1D and 1E do not resemble the same items; Why is the zoomed in insert in 1F (lower panel rotated= overall structure first, then then interaction with the glycerol and the substrate, followed by dimerization (which appears not to be that important for catalysis).

Include human MGL as a prototype MGL structure in the comparisons, e.g. there is wealth of data in the literature on the active site interactions (e.g. glutamate as glycerol stabilizing residue). Is human MGL considered as canonical monoacylglycerol lipase? The canonical MGL should be clearly defined when postulating the HBH topology as non-canonical prototype.

Figure 1A: what column was used for the SEC? the elution volumes are tiny, a molecular mass marker should be shown alongside. Have the monomer and dimer of the variant delta-lid been tested for activity separately?

Figure 2A and Figure 3A almost give the identical information. Figure 2A can be eliminated and the omit map be within Figure 3A .. Figure 2B is better placed alongside Figure 5.

“C18 activity”: The authors seem to oversize the C18 monoacyl ester intermediate. The structure is beautiful and it is amazing that it could be co-purified and co-crystallized. However, p-Nitrophenyl stearate (C18:0) 8% residual activity was reported already in 2009 (compared to 100% p-Nitrophenyl caprate (C10:0)), While this is clearly weaker than the activity toward a C14 (68%) or C16 (32%) substrate, the novelty of this intermediate being found, should not be overemphasized in the text. Maybe in vitro activities would show also these activities again, if different emulsifiers would have been tested. The authors themselves mention the low water-solubility of long-chain substrates. Different protocols for testing long-chain MG-substrates are available in the literature.

MGL-activity: Previous reports indicate activity also towards triacylglycerol. Although structurally, it might be difficult, experimental data from the previous report clearly indicate this activity. Was this tested in vitro with emulsified TG- substrates? Please comment.

High-resolution structure in the presence of an endogenous lipids: The complex-structure is indeed very nice, but the authors should note that there are other structures also in the PDB with bound long-chain ligands up to C18 either upon co-purification, co-crystallization or soaking e.g. PAM, OLA, OLB in fatty

acid binding proteins, 5BVS, 2FLJ, 5LXA or the mentioned monoglyceride lipase 4KE9 AlphaFold2 is mentioned. Assuming that during the training phase of AF2, the structure of Tth-MGL was not yet available, it would be interesting to show a brief overlay in cartoon representation of the experimental structure with the AF2 prediction would be interesting, since the new lid arrangement appears as an interesting challenge to the server. Why was the AF2 structure already predicted as a dimer - was this used as manual input (as indicated by the more than 500 amino acids)?

Significance: There is a wealth of data presented here, please highlight more in the conclusion why this protein is important, has potential in the view of the authors etc. Not much had been published on the lipase since its first discovery. Do the authors expect this to change with the detailed knowledge now available?

Minor concerns:

Can the authors compare the absolute activities of their purification and activities vs those of Royter et al, 2009?

Where are the positions of methylation?

Was the activity of the protein measured after the methylation process?

The resolution of the structures (Se-Met Tth-MGL 1.91) and PMSF-bound Tth-MGL (2.0) should be mentioned in the main text.

Supplementary Table S2 also indicates that stearic acid, and decan are present in the 'uncomplexed' structures. Where are they located in the structures ? Please indicate in a figure.

Why was the CD-Spectrum not quantitatively analyzed and compared to the 3D structure? All data should have been available.

Figure 1: Parts of the figure legend should better be positioned within the main text e.g. p12 starting from line 322)

Figure 3A, 3C: Where does the C18 chain reach the surface. 3A indicates the end of the chain behind the lid region, whereas it looks like the C18chain is in front of the cartoon in Fig 3C middle panel.

Figure 3 B and text – E43: acidic residues stabilizing a glycerol in the active site has been discussed in the MGL literature, e.g E53 from human MGL (Labar et al) – is this residue at a comparable structural position? How about E156 from MGL from *B. subtilis* – is this residue at a comparable position?

Figure 1E: what part of the substrate can be seen? Where would be the active site serine in the large cavity? The asterisk might be more confusing than helpful.

Figure 6A.

Include human monoacylglycerol lipase in the comparison

Please indicate the organism next to the protein name, e.g. 7Q4J monoacylglycerol lipase Tth, give at least one literature reference

7D79 and 5CML have been published and has a name other than NN

Figure 6B: Can you include in one figure the position of the ligands in the reference 7Q4J Tth-MGL and the other lipases (where known).

p9. line 269: the overall judgment with the previous thoughts in the literature appear somewhat harsh, lipase activity (at least towards medium chain length) has even been reported for *B. subtilis* lipase in 2001 where the lipase does not have a lid (see also Khan et al, 2017). Riegler-Berket et al specifically state that the cap-architecture in LipS, 3LLC, 4KE9 etc can be used to predict monoacylglycerol activity but "It should be noted, that this search is not exhaustive for lipases with activity towards MGs. This approach will not pick up enzymes with different fold or cap regions, even though they might be able to hydrolyze MGs (as is the case for pancreatic lipase related protein 2".. Along these lines, Tth-MGL would be another example. Bauer et al give a very nice overview of the modular structure of alpha/beta hydrolases. It would be great if the authors can make a brief reference to the lid vs cap naming convention they have suggested.

Kinetic data sometimes have decimal points, sometimes decimal commas.

Suppl Table S8: Please re-check the Assembly column for correctness of the biological assemblies.

POINT-TO-POINT RESPONSE ON REVIEWER COMMENTS

All replies inserted are in grey-blue both in the manuscript text and point-to-point response. Text quotations are marked. Similar referee comments have been cross-referenced.

In addition, we have shortened several section titles, to obey to the journal's regulations on section title length. For reasons of consistency, we also have introduced "MAG" as a new acronym for "monoacylglycerol".

For the referees' convenience, we are also including an updated list of figures and tables.

	Revised version	Previous version	Comments
Figures (Main)	1	1	Modified
	2	3	Modified
	3	4	Modified
	4	5	Panel A added
	5	6	Modified
Figures (Supplement)	S1	S1	
	S2	1F (main)	
	S3	2 (main), S4	Modified
	S4		New
	S5	S2	
	S6	S3	
	S7	S7	New
	S8	S5	
	S9	S6	
	S10	S7	
	S11	S8	
	S12	S9	
	S13		New
	S14	S10	
	S15		New
Tables (Supplement)	S1	S1	
	S2	S2	
	S3	S3	
	S4		New
	S5	S4	
	S6	S5	
	S7	S6	
	S8	S7	
	S9	S8	
	S10		New
	S11		New

Reviewer #1 (Remarks to the Author):

This manuscript has described a high resolution structure of monoacylglycerol lipase and the molecular basis for long-chain length substrate recognition have been highlighted. Some questions below are necessary to answer before considering published.

1、 In the "Extremophiles 13, 769-83 (2009) the lipase from *Thermoanaerobacter thermohydrosulfuricus* show substrate preference toward triacylglycerols with fatty acids in different chain lengths. Is the same enzyme described in this manuscript? The enzyme in this study is a

monoacylglycerol lipase not a TAG-hydrolyzing lipase, but the substrate characterization results have not seen.

The lipase described in this manuscript is identical to the one in Royter *et al.* (2009), in which the enzyme was described as: "... show activity toward a broad range of substrates, including triacylglycerols, monoacyl- glycerols, esters of secondary alcohols, and p-nitrophenyl esters." Indeed, Royter *et al.* (2009) measured turnover of triacylglycerols (TAGs), however, only after 24 hours incubation and quoting relative values only.

When repeating experiments under the experimental conditions established by Royter *et al.* (2009), we obtained 0.2 U/mg turnover of glyceryl trioctanoate (tricaprylin) as TAG model substrate over 24 hours. This is equivalent to 0.0014 U/mg over 10 minutes, which was chosen as time range in our turnover experiments (Figure 4B). Based on these data, we have concluded this activity to be insignificant and did not consider TAG turnover any further in this manuscript.

Furthermore, our structural work excludes any possibility to allow access to TAGs as two of three glycerol hydroxyl groups are blocked by *Tth* MAG lipase residues (see Figures 2B & 3).

For further details, see also reply to comment 19, referee 3.

2、 Reference 20(journal of Biological Chemistry 288.43 (2013): 31093-31104.) has described the crystal structure-substrate complex of a bacterial monoacylglycerol lipase, and also mentioned some key sites were importance for substrate binding. Please make some comments compared to your enzyme.

We have added the following sentence in the section

Tth MAG lipase is a minimal prototype for helix-β-hairpin-helix lid domain-containing esterases and lipases

"As key residues of the *Bacillus* sp. enzyme involved in regulation of MAG lipase turnover (Ile145, Glu156) are positioned on segments of the long HBH lid loop insert, which is missing in *Tth* MAG lipase (**Fig. 5b**), the precise mechanism of MAG lipase turnover regulation appears to be unrelated."

See also reply to comment 19, referee 3.

3、 Why monoacylglycerol lipase were selected for studying in this work? Is it important for industrial application or life science?

To strengthen the case for industrial applications we have added the following in the INTRODUCTION:

"This strain was originally isolated from Solar Lake in Israel, which has an extreme marine environment with a temperature range from 16 to 60°C and high level of salinity. These environmental conditions have given rise to complex biochemical phenomena that are linked to cycles of evaporation and infiltration of water external sources. Isolates from Solar Lake have shown remarkable biochemical processes related to degradation of starch, amylose, and pullulan ^{29,30}."

4、 The surface activation phenomenon are common for triacylglycerol lipase. Any evidences has been found in this study for the open and close conformation of lid domain?

Available evidence from our data for conformational changes of the lid domain in the presence and absence of the C18 reaction intermediate is described in detail in the section

Long chain monoacyl ester intermediate is held by a hydrophobic Tth MAG lipase lid tunnel

last paragraph, and illustrated in Figures cited in this paragraph. We hope that the referee is satisfied with the level of description provided.

5、 Glycerol products have been found this lipase structure. Does it has the relation with the glycerol releasing hole of monoacylglycerol lipase (Journal of Chemical Information and Modeling, 2021, 62(9): 2248-2256).

The glycerol releasing hole found in structures of the *Bacillus* sp. enzyme does not exist in *Tth* MAG lipase, as it does not contain a long HBH lid domain insert.

For further details, see our replies to comment 2, referee 1 and to comment 19, referee 3.

6、 In this work, HBH domain in lid of *Tth* MAG lipase have found to involve in long-chain length substrate recognition. But, lid domain in lipase have been confirmed for substrate recognition by intense studies. Although it is a nice work with fancy structural results, the finding here it is not so surprising.

We were actually surprised by the findings described in this contribution and particularly with the presence of the MAG reaction intermediate in the crystal structure. To the best of our knowledge, it is the first time that such reaction intermediate (as opposed to the reaction analogues) has been observed in a structure of this enzyme family.

See also reply to comment 3, referee 1.

Reviewer #2 (Remarks to the Author):

The Authors present nicely done high resolution crystal structures and provide a comprehensive analysis of the structures. Thorough evaluation of those interpretations was done providing enzymatic activity and enzyme kinetics data for mutants of residues deemed important. The Authors report an endogenous reaction intermediate which would be a significant observation. However there are some points unclear especially surrounding the intermediate that was found.

General comments

The Authors make the point that monoacylglycerol lipases are important for a number of industrial processes. However it would be nice to have more comment on how this particular enzyme is relevant in this sense. There are numerous enzymes of this class known that show high temperature stability with LipS even showing it's highest activity at 70C (doi:10.1371/journal.pone.0047665).

1. Given the impact factor of the journal it would be good to have more emphasis on how this new found structure and enzymatic activity is relevant. It would be good to have a stronger point on how this enzyme improves on the already known enzymes or how the findings stand out over previous studies. Alternatively a point could be made about the relevance of the enzyme for the survival of the host organism.

We have expanded the manuscript section

Tth MAG lipase is a minimal prototype for helix- β -hairpin-helix lid domain-containing esterases and lipases

by the following:

“Our data illustrate that a profound understanding of MAG lipase catalysis goes well beyond classical enzymology with well-defined active site / substrate relations. Knowledge on enzyme-specific contributions by lid structure and dynamics next and topping the active site are essential for our understanding of substrate specificity and possible future applications on lipase design with further improved turnover properties. Our structure data also reveal the importance on the way how the glyceride backbone of lipase acyl substrates is bound by specific active site residues, ultimately defining its activity profile as MAG lipase, which is not permissive to act as a diacylglyceride or triacylglyceride lipase.”

We also refer to our reply to comment 3, referee 1.

2. The Authors report a reaction intermediate, it is unclear what exactly is meant by this. Do the authors claim to see the short lived tetrahedral intermediate formed during the enzymatic reaction?

In this case it would be good to have comment on why this intermediate is supposedly stable in this particular case and why the reaction would not go to its end. All that would be needed is to finish the reaction is water which should be abundantly available in a protein crystal available.

It should also be made clearer what reaction intermediate the authors are proposing, this could be marked in Figure 4. If this indeed represents an endogenous intercommunication this is very significant, however there is not much evidence provided for this and liquid chromatography mass spectroscopy seems to indicate the presence of uncleaved substrate.

We thank the reviewer for this very valuable comment and further related comments below.

What we observe in the *Tth* MAG lipase structure is due to crystallization conditions and does not present a catalysis intermediate observed in solution, where there is abundant access of water as a requirement of MAG lipase hydrolysis. Reinvestigation of the *Tth* structure shows that there are no water molecules within 7 Å distance to the C1 atom of the C18 MAG reaction intermediate, which is targeted by nucleophilic attack during hydrolysis. Hence, we have concluded that the lack of water molecules near the active site reaction center may be a possible reason why *Tth* MAG lipase stops prior to the involvement of water required for MAG hydrolysis under the chosen crystallization conditions.

Based on these observations we have added the following in the manuscript text, section

Long chain monoacyl ester intermediate is held by a hydrophobic Tth MAG lipase lid tunnel:

“Based on these data, we modeled a monostearate (C18) acyl ligand into the electron density of the active *Tth* MAG lipase structure (**Fig. 2a & Supplementary Fig. S3a**). Since we observed significant connecting density between the terminal C1 carbon of the C18 monoacyl ligand and both the γ -hydroxyl oxygen of Ser113 and the O1 oxygen of the glycerol moiety, unrestrained refinement produced distances between these atoms in the range of 2.2-2.6 Å. The resulting model, which was confirmed by an omit electron density map, supports the presence of a loosely coordinated tetrahedral MAG ester intermediate^{14,33,34}, preceding water-mediated ester hydrolysis (**Fig. 2a-b & 3, Supplementary Fig. S3-S4**). Investigation of the structural neighborhood of the C1 carbon atom of the C18 ligands in both *Tth* protein chains found in the crystal structure, revealed a complete lack of any solvent molecules within a 7 Å radius, suggesting that under these crystallization conditions the reaction stops prior to solvent-mediated hydrolysis.”

We have reassessed all statements throughout the manuscript to ensure that the term “intermediate” is only used in the context of our structural observations. Where the term “catalysis intermediate” was used we have replaced it by “reaction intermediate”, to express that our structure-based observations are due to one-time termination of a specific reaction step in *Tth* MAG lipase substrate turnover rather than due to *Tth* MAG lipase ongoing catalysis. To ensure that the term is appropriately used in the

title of the manuscript as well, we have modified it to: **“Discovery of a non-canonical prototype long-chain monoacylglycerol lipase through a structure-based endogenous reaction intermediate complex”**.

Specific comments

3. Page 2 The Authors make the point that this structure would not have been accessible normally due to low solubility of the ligand However there are structures published with high chain length substrates (PDB 4KE9, 4KE8) indicating that the substrates can be made soluble enough to deliver them to the protein.

PDB codes 4KE8 and 4KE9 contain C14 and C16 phosphonate analogues, respectively, which lead to covalently bound irreversible analogue intermediate complexes. The respective analogues were solubilized in ethanol prior crystallization to increase their solubility (for details see Rengachari *et al.*, 2013). The approach used to determine the respective structures is complementary to ours and equally valid. We have added the following in the section

Long chain MAG ester intermediate is held by a hydrophobic Tth MAG lipase lid tunnel

“Our structural findings on the presence of a C18 MAG intermediate are also in agreement with previously used MAG intermediate mimicking analogues ¹² (**Supplementary Fig. S4**).”

Supplementary Figure S4 shows a superposition of our *Tth* MAGb lipase structure onto the C14 phosphonate analogue complex deposited in PDB code 4KE8, demonstrating resemblance of MAG reaction intermediate geometry. In contrast, the distal groups (C18/C14 and glycerol) deviate significantly in the two superimposed structures.

4. Page 3 The Authors make the point that the limited solubility of the substrates make previously reported substrate profiles unreliable. However it is not mentioned how this is not a problem in the study presented here. The solubility of the p-nitro-phenol substrate used by the authors will also decrease with increasing carbon chain length, this is even described in the supplementary section. It would be good to have an explanation why this is considered a problem in previous studies but not in this one.

The problem of long chain-substrate solubility has been as relevant in past studies as in this contribution for *in vitro* studies using purified protein. Hence, essential additional approaches in this manuscript are in the use of structural data with a C18 intermediate from the expression host as well as the cell extract analysis, presented in the last paragraph of the section *Mechanism for variable length monoacyl ester turnover by Tth MAG lipase*. The combined use of structural data, *in vitro* biochemical data and cell extract analysis, to the best of our knowledge, is without precedence in the relevant literature.

5. Page 3 The experimental details described seem a bit slim. It is usual to mention crystallization conditions and at least the final purification buffers. It doesn't seem likely to me the work presented could be reproduced with the experimental details given.

We are wondering whether the referee only looked at the experimental summary at the beginning of the document or also at the complete Methods sections at the end of the document, where a detailed description is provided? The Methods Summary section, which was included in the initial manuscript version, has been removed in the revised version, to follow the journal's format regulations.

To improve the structure and content of the relevant METHODS section, we expanded the section *Protein expression and purification*. We have also restructured the previous section *Tth MAG lipase crystallization and analysis of crystal content* into three separate sections entitled *Tth MAG lipase*

reductive methylation, Tth MAG lipase crystallization and analysis of crystal content. The last section follows the section *X-ray data collection, structure solution, and validation.* As we have included an additional *Tth MAG lipase* structure to support our findings, we have rewritten the section *Tth MAG lipase crystallization.* All changes and additions in the sections are highlighted by blue color.

6. The Authors also make a point of seeing glycerol in their binding pocket even in the presence of PMSF (Page 5). Glycerol is a very common additive during protein purification and crystallization and one of the most popular cryo-protectant during freezing of protein crystals. Without more experimental details it is difficult to judge whether this glycerol could come from the host organism or the sample preparation.

Our observation of continuous density between γ -hydroxyl oxygen of Ser113 and the O1 oxygen of the glycerol moiety in the structure of *Tth MAG lipase* in complex with the C18 glyceride reaction intermediate is suggestive that this results from glycerol monostearate, as further evidenced by LC-MS (Supplementary Figure S3). However, for the PMS-inhibited *Tth MAG lipase* structure the argument about an unknown original of glycerol in the active site is valid. To address this question, we have determined an additional structure of non-methylated *Tth MAG lipase* where no glycerol was used as cryo-protectant (Supplementary Figure S7). As this structure is inferior in terms of resolution (2.42 Å) to the structure of methylated *Tth MAG lipase* (1.91 Å resolution), we use the structure solely for the purpose of supporting the origin of glycerol in the *Tth MAG lipase* active site independent of specific crystallization and cryo-protectant conditions. Crystallization and details of the determination and refinement of this structure have been added in the METHODS section.

We have added the following in the section

Long chain monoacyl ester intermediate is held by a hydrophobic Tth MAG lipase lid tunnel:

“To rule out that the presence of glycerol in the *Tth MAG lipase* structure is due to the addition of glycerol in the cryo-protectant buffer, we determined an additional *Tth MAG lipase* structure without using glycerol in any sample preparation step and cryo-protection prior to X-ray data collection (**Supplementary Table S2**). In this structure, glycerol is found in a literally identical active site position in five out of six protein chains of the respective asymmetric unit, confirming its origin to be independent from a specific crystallization and cryo-protection protocol (**Supplementary Fig. S7**).”

Supplementary Figure S7 illustrates a position of glycerol molecules in the active sites of the two protomers of one of the *Tth MAG lipase* dimers, which are identical to those found in the *Tth MAG lipase* C18-acylglycerol and PMS complexes.

7. More details on how the enzymatic assays were done would also be good to better interpret the results.

Same as for reply to comment 5, referee 3: we are wondering whether the referee only looked at the experimental summary at the beginning of the document or also at the complete METHODS section at the end of the document, where a detailed description is provided?

We have added more details in the METHODS section *In vitro assay to determine lipolytic enzyme activity.* All changes and expansions are highlighted in blue.

8. On Page 5 The authors state, they find a single peak for glycerol-monostearate doing mass spectrometry, however in the same sentence they interpret this in a manner that product cannot be released. This seems somewhat contradictory a glycerol-monostearate should be a substrate to a monoacylglycerol lipases not product. The Authors also report a reaction intermediate in this manuscript, it seems their mass spectroscopy findings might contradict this. If the fatty acid is in one of the intermediate states it should be contently linked to the enzyme (Figure 4), does that not

contradict the mass spectrometry findings?

We have improved text and figures to unambiguously illustrate the nature of the structure-based C18 MAG reaction intermediate, by consistently indicating the two transient bonds between the C1 carbon of the C18 acyl moiety and the O1 oxygen of the glycerol moiety as well as the O γ (OG) oxygen of the active site residue Ser113. For further details on text, see reply to comment 2, reviewer 2.

Supplementary Figure S3A has been modified to demonstrate continuous density for these transient bonds, based on an *2Fo-Fc* omit map of the refined model of the MAG reaction intermediate. Figures 2A and 3 have also been modified accordingly. Supplementary Figures S4 and S15 were added.

A most plausible model to explain the different findings by LC-MS experiments and X-ray crystallography is by the established preference of lipases for esterification as opposed to hydrolysis in the presence of organic solvents, which were used to prepare dissolved *Tth* MAG lipase crystals for LC-MS analysis. We have added the following in the text section

Long chain MAG ester intermediate is held by a hydrophobic Tth MAG lipase lid tunnel.

“As *Tth* MAG lipase is exposed to an acidic water/organic solvent mixture in LC-MS experiments after its re-solubilization from crystals favoring esterification³⁵, our most plausible explanation is a reverse process from the C18 reaction intermediate observed in the crystal structure to glycerol monostearate substrate detected in LC-MS experiments.”

9. On Page 7 The Authors mention an unexpected gain of function for a Glu43 variant. This unexpected behavior might be a result of the *p*-nitrophenol leaving group being handled differently than a glycerol leaving group in the crystal structure. It would be interesting to see the behavior of this variant in an assay using a glycerol ester as substrate.

We have tested alternative *in vitro* assays but unfortunately those have by not been as reliable as the mono *p*-nitrophenyl ester assay. In part because of this, we have established a cell extract assay (for details see Figure 4E). In agreement with the *p*-nitrophenyl ester assay (Figure 4B), our cell extract assay reveals significant gain-of-function for the *Tth* MAG lipase E43A mutant. Should the gain-of-function observed in the *p*-nitrophenyl ester assay have been due to the presence of the *p*-nitrophenyl leaving group, it would have been unlikely to observe a related behavior of this mutant in the cell extract assay.

10. Figure 2 The Authors claim to have found a reaction intermediate, however in their structure it seems there is a covalent bond missing between the glycerol and the fatty acid chain.

It should be clarified why this bond is missing.

If this is supposed to either be the substrate or the intermediate, this bond should be considered during ligand placement and refinement since it might put a restraint on the geometry of the substrate that is missing without the bond.

Additionally it would be good to have more information about the geometry, depending on what intermediate it is there should be specific angles around the alpha carbon of the fatty acid moiety.

See reply to comment 8, reviewer 2.

Relevant geometry values of the C18 MAG intermediate are listed in the additional Supplementary Table S11, which complements the additional Supplementary Figure S4.

11 One concern I have is that using this structure as evidence for the intermediate seems insufficient, there needs to be experimental proof that there is a bond between the substrate and the enzyme, especially since a peak for the substrate in mass spectrometry already suggests that this might just be bound substrate that is not contently linked.

See replies to previous comments of this referee.

As our observations are most likely due to a specific environment of the *Tth* MAG lipase active site under protein expression, purification and subsequent crystallization conditions of the methylated version of the enzyme, there are no options to prove such state beyond the specific conditions in a crystalline environment. Although the observation of the structure-based C18 MAG intermediate was very helpful for the main topic of this contribution, the discovery of this enzyme to act as a long-chain MAG lipase, biochemically unproven details of the underlying catalysis mechanism have no impact on the enzyme's functional characterization.

Reviewer #3 (Remarks to the Author):

This interesting study presents the structure of an enzyme from the thermophilic anaerobic bacterium *Thermoanaerobacter thermohydrosulfuricus* (Tth) that has first been described in 2009 as LipTth and according to the amino acid sequence and substrate activity against “a broad range of substrates, including triacylglycerols, monoacylglycerols, esters of secondary alcohols, and p-nitrophenyl esters” has been assigned as a lipase. Medium chain Length (C8-C14) had been reported as the preferred chain lengths for p-nitrophenyl ester substrates. LipTth had also been reported to be highly S-stereoselective towards esters of secondary alcohols with high robustness with respect to temperature and different solvents. These features render the lipase potentially interesting for biotechnological studies and the aim of the present study is to understand enzymes activity, structure and substrate promiscuity using X-ray crystallography combined with biochemical and biophysical studies of wt protein and mutations thereof. In this study, the protein is termed Tth-MAG lipase. Crystal structures were determined in presence of PMSF, and in the absence of PMSF a C18 monoacyl ester intermediate was observed in the active site tunnel, showing a large substrate spectrum. Also of interest is, that crystals were only obtained after methylation and the structures were solved with Se-Met Tth MAG lipase.

Major concerns:

1. Overall: The data in the manuscript are really interesting and detailed for the MAG lipase community. However the manuscript is difficult to read in some parts and might benefit from a restructuring e.g. aid the reader in the orientation of the structures (1D, E and F all have different orientations); the colors in 1D and 1E do not resemble the same items; Why is the zoomed in insert in 1F (lower panel rotated= overall structure first, then then interaction with the glycerol and the substrate, followed by dimerization (which appears not to be that important for catalysis).

The color coding in Figure 1E has been corrected, as indeed the color codes for the Catalytic domain (CD) and lid domain were inverted with respect to the previous panels in the original figure. To safeguard that there is no confusion of using strong colors for different purposes in panels D and E, we have introduced a second color for the CD of the second protomer (pale yellow). Since the lid domains are well separated in this presentation, we have used the same color (light pink) for illustrating the lid domains of both protomers.

According to the referee's suggestions we have moved panel F of Figure 1 (old) into the supplement, by generating an additional Supplementary Figure S2.

2. Include human MAG lipase as a prototype MAG lipase structure in the comparisons, e.g. there is wealth of data in the literature on the active site interactions (e.g. glutamate as glycerol stabilizing residue). Is human MAG lipase considered as canonical monoacylglycerol lipase? The canonical MAG lipase should be clearly defined when postulating the HBH topology as non-canonical prototype.

We have added the following sentence in section

Tth MAG lipase is a minimal prototype for helix- β -hairpin-helix lid domain-containing esterases and lipases:

“This motif is not present in most previously characterized MAG lipases that comprise lid domains with one or more helices without a β -sheet inserted²², we define here as canonical. These include human MAG lipase, which has been used as model in various studies^{15,39} (Fig. 5a).”

3. Figure 1A: what column was used for the SEC? the elution volumes are tiny, a molecular mass marker should be shown alongside.

We have added the following sentence in the METHODS section *Protein expression and purification*:

“For SEC profile analysis, a Superdex 75 increase 3.2/300 column (Cytiva) was used.”

4. Have the monomer and dimer of the variant delta-lid been tested for activity separately?

We have tested the activities of the two peaks to the extent significant, presenting dimeric and monomeric *Tth* MAG lipase, separately, but did not detect any significant differences. The values quoted in this contribution are for dimeric *Tth* MAG lipase.

5. Figure 2A and Figure 3A almost give the identical information. Figure 2A can be eliminated and the omit map be within Figure 3A .. Figure 2B is better placed alongside Figure 5.

Figure 2A has become part of Supplementary Figure S4 as requested. Showing the density in addition in Figure 2A would obscure other messages of this figure. Showing the density of the MAG intermediate in its improved form, in our view, is still a key message of this contribution, and the manuscript would win if this returns back into the main body of this contribution.

Figure 2B has been moved to Figure 4 as suggested.

6. “C18 activity”: The authors seem to oversize the C18 monoacyl ester intermediate. The structure is beautiful and it is amazing that it could be co-purified and co-crystallized. However, p-Nitrophenyl stearate (C18:0) 8% residual activity was reported already in 2009 (compared to 100% p-Nitrophenyl caprate (C10:0)), While this is clearly weaker than the activity toward a C14 (68%) or C16 (32%) substrate, the novelty of this intermediate being found, should not be overemphasized in the text. Maybe *in vitro* activities would show also these activities again, if different emulsifiers would have been tested. The authors themselves mention the low water-solubility of long-chain substrates.
Different protocols for testing long-chain MG-substrates are available in the literature.

We have tested several variations of the *in vitro* lipase assay presented in this publication. The effect of different emulsifiers on enzyme activity was also evaluated, and gum arabicum was identified as best option during the development of the assay presented in this contribution. Other tests included the evaluation of enzyme delipidation protocols prior to assaying enzyme activity. We would like to stress that, in addition to the structural data presented, cell extract analysis (Figure 4E) also clearly indicated a preference for long MAGs.

7. MAG lipase-activity: Previous reports indicate activity also towards triacylglycerol. Although structurally, it might be difficult, experimental data from the previous report clearly indicate this activity. Was this tested *in vitro* with emulsified TG- substrates? Please comment.

See reply to referee 1, comment 1.

8. High-resolution structure in the presence of a endogenous lipids: The complex-structure is indeed

very nice, but the authors should note that there are other structures also in the PDB with bound long-chain ligands up to C18 either upon co-purification, co-crystallization or soaking e.g. PAM, OLA, OLB in fatty acid binding proteins, 5BVS, 2FLJ, 5LXA or the mentioned monoglyceride lipase 4KE9

We added the following sentence in the INTRODUCTION:

“Although protein structures in the presence of large hydrophobic substrates or ligands have been published, often with a requirement of applying target-tailored solubilization protocols⁹⁻¹², they still remain exceptional and hence available structure repositories remain mostly populated with protein structures of soluble, small ligands.”

However, none of the ligands found in these structures have been extracted from the expression host.

For further details, see also reply to comment 3, referee 2.

9. AlphaFold2 is mentioned. Assuming that during the training phase of AF2, the structure of *Tth*-MAG lipase was not yet available, it would be interesting to show a brief overlay in cartoon representation of the experimental structure with the AF2 prediction would be interesting, since the new lid arrangement appears as an interesting challenge to the server. Why was the AF2 structure already predicted as a dimer - was this used as manual input (as indicated by the more than 500 amino acids)?

We have added the following text in the METHODS section:

“AlphaFold2 (<https://colab.research.google.com/>) was used for modeling the *Tth* MAG lipase structure from the sequence, by forcing a dimeric arrangement. The resulting models superimposed onto the *Tth* MAG lipase crystal structures with a root mean square deviation of about 0.45 Å.”

10. Significance: There is a wealth of data presented here, please highlight more in the conclusion why this protein is important, has potential in the view of the authors etc. Not much had been published on the lipase since its first discovery. Do the authors expect this to change with the detailed knowledge now available?

See reply to comment 1, referee 2.

Minor concerns:

11. Can the authors compare the absolute activities of their purification and activities vs those of Royter et al, 2009?

Unfortunately, all activity assay results by Royter *et al.* (2009) were expressed in percentages. The only exception we found was in the purification protocol (Table 2), where absolute values were presented aiming to demonstrate progress of protein purity. According to the data, maximum activity of purified protein against C16 substrate was 12.14 U/mg. Along the experimental conditions cited, purified protein contained 1% DMSO and the substrates added were in 100% DMSO, leading to a final DMSO concentration of about 6% in the reaction buffer. In addition, activity assays were carried out at an elevated temperature of 70 °C, to mimic close to native conditions of the organism (*Thermoanaerobacter thermohydrosulfuricus*), from where the gene coding for this lipase was taken.

For our measurements and also structural work aiming for a mechanistic understanding, the conditions originally applied, both for protein purification and activity assays, were not suitable to allow generating quantitative and reproduceable data. Hence, our data obtained were in the absence of the addition of further organic solvents such as DMSO leading to solubilization of substrates with little solubility and at a lower temperature (40 °C). Hence, we only found residual activity for C16

substrate (0.04 U/mg) (Figure 5B). Taking into account these differences of experimental conditions, in our view any further comparison is not relevant.

For further details, see also reply to comment 6, referee 3.

12. Where are the positions of methylation?

To address this question, we have added a new Supplementary Figure S13 and Supplementary Table S10.

13. Was the activity of the protein measured after the methylation process?

We have measured *Tth* MAG lipase activity after reductive methylation, indicating that this has no significant negative effect on catalytic activity. We have added the following in the METHODS section

Tth MAG lipase reductive methylation:

“Catalytic activity against C8 *p*-nitrophenyl monoacyl ester using the assay as described in section “*In vitro* assay to determine lipolytic enzyme activity” was close to non-methylated *Tth* MAG lipase (2,3 U mg⁻¹, cf. **Figure 4B, left panel, and Supplementary Table S3**).”

14. The resolution of the structures (Se-Met Tth-MAG lipase 1.91) and PMSF-bound Tth-MAG lipase (2.0) should be mentioned in the main text.

The values have been added in the text.

15. Supplementary Table S2 also indicates that stearic acid, and decan are present in the ‘uncomplexed’ structures. Where are they located in the structures ? Please indicate in a figure.

We added a new Figure (Supplementary Figure S15) to illustrate the positions of other non-validated ligands.

16. Why was the CD-Spectrum not quantitatively analyzed and compared to the 3D structure? All data should have been available.

A new Supplementary Table S4 and the following text has been added including a description in the methods:

“The estimation of the secondary structure contents was calculated using CONTIN⁵⁵.”

17. Figure 1: Parts of the figure legend should better be positioned within the main text e.g. p12 starting from line 322)

In the submitted version all figures including the legends were positioned at the end of the manuscript. We fix this according to the publisher’s instructions.

18. Figure 3A, 3C: Where does the C18 chain reach the surface. 3A indicates the end of the chain behind the lid region, whereas it looks like the C18chain is in front of the cartoon in Fig 3C middle panel.

The distal end of C18 ligand is indeed exposed to the surface, as also indicated in the text in section

Long chain monoacyl ester intermediate is held by a hydrophobic Tth MAG lipase lid tunnel:

“The distal part of the C18 MAG ester intermediate involving carbon atoms C11 to C18 is held by a hydrophobic tunnel across the *Tth* MAG lipase HBH lid domain (**Fig. 2a & c**). In the C18 MAG intermediate bound *Tth* MAG lipase structure the lid tunnel is open at both ends (**Fig. 2e**). Whereas the proximal opening towards the *Tth* MAG lipase active site can be viewed as an extension of the active site area (**Fig. 2a**), the distal lid tunnel exit is exposed and defined by a ring-like structure of side chains from hydrophobic residues (**Fig. 2d-e**).”

As this finding is indeed not obvious from panel A, we are especially addressing this point in panels C (left) and E (left).

19. Figure 3 B and text – E43: acidic residues stabilizing a glycerol in the active site has been discussed in the MAG lipase literature, e.g E53 from human MAG lipase (Labar et al) – is this residue at a comparable structural position? How about E156 from MAG lipase from *B. subtilis* – is this residue at a comparable position?

We thank the reviewer for noting this. Indeed, the side chain of Glu53 of human MAG lipase is similarly involved in interacting with the two other glycerol oxygens as observed in the *Tth* MAG lipase presented in this contribution. One could speculate that this demonstrates part of a key mechanism rendering both human MAG lipase and *Tth* MAG lipase to be specific for MAG substrates. However, Glu43 from *Tth* MAG lipase and Glu53 from human MAG lipase are in different sequence positions, and hence one could argue to support an argument of convergent evolution. As indicated in the sequence alignment of Supplementary Figure S5, the position of Glu43 in *Tth* MAG lipase is part of a loop segment, which is not conserved both in length and sequence. The position Glu53 of human MAG lipase is equivalent to Gly39 of *Tth* MAG lipase, which is not conserved either. As human MAG lipase is not a lipase with an HBH lid segment, which is the focus of this contribution, we feel that including such comparison would be beyond the scope of this paper.

Concerning the role of Glu156 from the *Bacillus* sp. MAG lipase, which does contain an HBH lid domain, we refer to our reply on comment 2, referee 1.

20. Figure 1E: what part of the substrate can be seen? Where would be the active site serine in the large cavity? The asterisk might be more confusing than helpful.

The following has been added in the figure legend for clarification:

“... and two asterisks in red, indicating the two active sites. In this presentation, the proximal part of the C18 MAG ligands is visible (*cf.* **Fig. 2**). The orientation of *Tth* MAG lipase protomer A is rotated by about 90 degrees around a vertical axis with respect to the *Tth* MAG lipase monomer shown in panel d, as indicated.”

Unfortunately, the active site Ser113, which is used as active site marker in other figures, is not visible in this presentation. Hence, we are indicating the active site with an asterisk close to Ser113. If the referee prefers to remove the asterisk, of course this can be done.

21. Figure 6A.

Include human monoacylglycerol lipase in the comparison

Please indicate the organism next to the protein name, e.g. 7Q4J monoacylglycerol lipase *Tth*, give at least one literature reference

7D79 and 5CML have been published and has a name other than NN

Figure 6B: Can you include in one figure the position of the ligands in the reference 7Q4J *Tth*-MAG lipase and the other lipases (where known).

In the revised Figure 5A a representative for human MAG lipase has been included. The organism names are also added. Relevant literature references and the organism names are also listed in

Supplementary Table S9. A reference to this table has been added in the figure caption. The functional names for 7D79 and 5CML as extracted from the literature have been added as well. In all structures shown in panel B, active ligands have been added and explained in the figure caption by the following addition:

“Active site ligands are shown in sphere presentation in atom-specific colors (carbon, green; oxygen, red; nitrogen, blue). Only in *Tth* MAG lipase, the C18 ligand crosses the lid domain via a hydrophobic tunnel. The ligands in 3LLC (tetraethylene glycol) and 4KE8 (tetradecyl hydrogen (R)-(3-azidopropyl)phosphonate, chain A) are in similar positions next to the active site but they are too short to cross the respective lid domains (**Supplementary Fig. S4a**).”

22. p9. line 269: the overall judgment with the previous thoughts in the literature appear somewhat harsh, lipase activity (at least towards medium chain length) has even been reported for *B. subtilis* lipase in 2001 where the lipase does not have a lid (see also Khan et al, 2017). Riegler-Berket et al specifically state that the cap-architecture in LipS, 3LLC, 4KE9 etc can be used to predict monoacylglycerol activity but “It should be noted, that this search is not exhaustive for lipases with activity towards MGs. This approach will not pick up enzymes with different fold or cap regions, even though they might be able to hydrolyze MGs (as is the case for pancreatic lipase related protein 2”.. Along these lines, *Tth*-MAG lipase would be another example. Bauer et al give a very nice overview of the modular structure of alpha/beta hydrolases. It would be great if the authors can make a brief reference to the lid vs cap naming convention they have suggested.

We have removed the following:

“...unlike previous thoughts in the literature ^{18,21,35}”

from the text, as this is no intention to devalue previous work and any further comparative discussion probably would be better placed in a future review. We have added the following when we mention “lid” the first time in the text:

“Interfacial activation of lipases is thought to require an additional α -helical domain referred to as “lid”, alternatively also named “cap”, which contains several hydrophilic residues in the vicinity of the active site ²².”

23. Kinetic data sometimes have decimal points, sometimes decimal commas.
Suppl Table S8: Please re-check the Assembly column for correctness of the biological assemblies.

This has been corrected. Decimals of all kinetic data are now shown by decimal points.

REVIEWER COMMENTS

Reviewer #2 (Remarks to the Author):

The Authors rearranged the Methods in a way to make it less confusing and added experimental details. They show additional evidence to solidify their claim to observe a reaction intermediate. They have improved text and figure to make it clear that the bond between the glycerol moiety and the fatty acid moiety exists and show electron density linking those moieties together.

The explanation how this intermediate comes to be still a little bit doubtful to me. It seems unrealistic to me that this would come from a lack of water, during the purification and crystallization process there is ample water available and enough time for the enzyme to turn over the last substrate left over from the host organism. Additionally protein crystals are usually built up by 40-50% water.

The Authors answered most of the questions and made efforts to rearrange the manuscript and change figures to make things more understandable.

One question however seems to have been misunderstood. I tried here to clarify my confusion about the enzymatic activity assays.

Reviewer 2 Question 9:

The question might have been formulated in a confusing way. The question was not about an in vivo assay or biological relevance but about the fact that the conclusions are made for MAG substrates while p-nitrophenoyl esters are used in the assay. MAGs have a glycerol as head group while p-nitrophenoyl esters don't contain glycerol but a nitrophenyl group instead. The question was if the gain of function, that seemed to be unexpected could have come from the fact that the structure interpretation was done for a glycerol head group while the assay uses nitrophenol. Those two head groups might be handled different by the enzyme.

In the method section "cleavage of p-nitrophenoyl MAG esters ..." the Authors have reference 28. However to my understanding this paper uses mostly p-nitrophenoyl FA esters as substrate. There is a list of other substrate tested however it seems p-nitrophenyl palmitate was the major substrate in most of the experiments. This particular assay is not an MAG hydrolase assay since there is no glycerol moiety. Could the authors clarify whether an actual p-nitrophenoyl MAG ester was used containing both a glycerol moiety and a p-nitrophenyl group or if the p-nitrophenyl FA ester was used as representative for MAGs?

p-nitrophenyl assays is a common assay for lipases however MAGs and p-nitrophenoyl esters have different head groups. p-nitrophenyl are a more unspecific substrate for fatty acid ester hydrolases and enzymes can have high activities for p-nitrophenyl-FA ester while low activity for MAGs. This might lead to the unexpected results.

My Question basically was if there is a reason why the substrate used was not actual MAGs while measuring released glycerol or fatty acids as read out?

Are all the assays in the manuscript that refer to MAG activity done with p-nitrophenyl-FA-esters? If this is the case it is actually possible the the enzyme has very low MAG activity. That could also explain why intact MAG subtrates was found in mass spectroscopy of the crystals.

Please clarify what substrate was used (CAS number if necessary), it seems to me the Authors might have tested for the wrong substrate for the conclusions they are trying to make.

Reviewer #3 (Remarks to the Author):

The authors have significantly improved the clarity and presentation of the manuscript in the revised version.

The science presented is well supported by experimental data, yet some loose ends remain (which might be the case in any manuscript). The term ester-intermediate is somewhat unusual.

Minor comment: Fig 1: D and M are not in the figure, yet described in the figure legend, please correct

It is interesting that the authors bring estrification vs hydrolytic reaction of the lipase even more into the focus. As the authors correctly address, it is not entirely clear why this product/substrate has been identified in the active site even upon isolation.

In conclusion, the study on the monoacylglycerol-lipase activity is very nice and comprehensive. It would have been beneficial if more specific ideas on the use of enzymes from Lake Solar were includede, ref. 29 and 30 date from 1990 and 1977, respectively. This could widen the interest in the large piece of work broader than the MAGL community.

POINT-TO-POINT RESPONSE ON REVIEWER COMMENTS

All replies inserted are in grey-blue both in the manuscript text and point-to-point response.

Reviewer #2 (Remarks to the Author):

The Authors rearranged the Methods in a way to make it less confusing and added experimental details.

They show additional evidence to solidify their claim to observe a reaction intermediate. They have improved text and figure to make it clear that the bond between the glycerol moiety and the fatty acid moiety exists and show electron density linking those moieties together.

The explanation how this intermediate comes to be still a little bit doubtful to me. It seems unrealistic to me that this would come from a lack of water, during the purification and crystallization process there is ample water available and enough time for the enzyme to turn over the last substrate left over from the host organism. Additionally protein crystals are usually built up by 40-50% water.

We have reanalyzed our structural data for three alternative models, as illustrated in the Figure below.

Figure caption: We have refined three closely related conformational states, using $2Fo-Fc$ omit maps (light blue) at 1σ level and a negative $Fo-Fc$ difference map (red) at 2.5σ level. Only density near *Th* MAG lipase residues 112-114 and 233 and bound C18-monoglyceride is shown for the sake of clarity. Residue and ligand colors are as in other figures of the manuscript. Transient bonds are indicated by dashed lines in yellow.

In the **loose C18 reaction intermediate model (left panel)**, densities bridging C1(C18)–O1(GOL) and C1(C18)–OG(Ser113) are interpreted as transient bonds with increased length of 2.2- 2.6 Å (Supplementary Table S11). In addition, there is only residual negative difference density next to this site. This model is presented in the manuscript as the one most closely fitting experimental electron density.

Interpretation of the density as a **tight C18 reaction intermediate (central panel)** requires assignment of covalent bonds to C1(C18)–O1(GOL) and C1(C18)-OG(Ser113), similar to previous structural observations of another MAG lipase in presence of a covalently bound intermediate mimic (4KE7). This model allows interpretation of densities bridging between C1(C18)-O1(GOL) and C1(C18)-OG(Ser113). In contrast to the loose reaction intermediate (left), this interpretation generated significant negative difference density close to these bonds, hence this interpretation has been less favored by us.

Modeling of a **C18 MAG substrate complex (right panel)** does not properly address the density connecting C1(C18)-OG(Ser113), as in this model there is no bond between these two atoms. In addition, there is still significant negative difference density next to the bond C1(C18)-O1(GOL), for

the same reason as observed when interpreting the density as tight reaction intermediate (central panel). Hence, this interpretation has also been less preferred by us.

All three interpretations are so minute in difference that they have no significant impact on the overall structural refinement statistics (Supplementary Table S2). Nevertheless, as we can observe significant differences by carefully scrutinizing available density at the given high resolution (1.91 Å), we believe it is worth stating this in the manuscript, although the definition of the precise state had no impact on the design of all subsequent functional experiments, ultimately leading to the discovery of the enzyme as a highly specific long-chain MAG lipase. We have added the following sentence in the manuscript at the end of the presentation of the structural data: *“All subsequent experiments were designed by interpreting our structural and mass spectrometry-based findings as C18 MAG.”*

Should there be a wish by the editor or referees to include the comparison illustrated above, this analysis could be added by another Supplementary Figure.

We also agree with the referee’s comments that speculating on what may have happened during the crystallization process does not provide added value and hence we have removed any indication, which could be interpreted as such. Additional analysis with available MAG lipase structures in absence or presence of tetrahedral mimics (4KEA, 4KE7, 4KE8) shows, however, that there is a conserved water molecule next to the catalytic triad residue H233 (His 226 in *Bacillus* lipase) required for hydrolysis (Rengachari *et al.*, 2013). Such water molecule is missing in our structure. In this revised version, this is mentioned as a structure-based observation only. We have replaced the previous text by the following sentence: *“We also noticed that a conserved solvent molecule required for ester hydrolysis next to the active site triad residue H233 detected in those structures [Rengachari, 2013 #190] is missing in our Tth MAG reaction ligand complex.”*

The Authors answered most of the questions and made efforts to rearrange the manuscript and change figures to make things more understandable.

One question however seems to have been misunderstood. I tried here to clarify my confusion about the enzymatic activity assays.

Reviewer 2 Question 9:

The question might have been formulated in a confusing way. The question was not about an in vivo assay or biological relevance but about the fact that the conclusions are made for MAG substrates while p-nitrophenyl esters are used in the assay. MAGs have a glycerol as head group while p-nitrophenyl esters don’t contain glycerol but a nitrophenyl group instead. The question was if the gain of function, that seemed to be unexpected could have come from the fact that the structure interpretation was done for a glycerol head group while the assay uses nitrophenol. Those two head groups might be handled different by the enzyme.

In the method section “cleavage of p-nitrophenyl MAG esters ...” the Authors have reference 28. However to my understanding this paper uses mostly p-nitrophenyl FA esters as substrate. There is a list of other substrate tested however it seems p-nitrophenyl palmitate was the major substrate in most of the experiments. This particular assay is not an MAG hydrolase assay since there is no glycerol moiety.

Could the authors clarify whether an actual p-nitrophenyl MAG ester was used containing both a glycerol moiety and a p-nitrophenyl group or if the p-nitrophenyl FA ester was used as representative for MAGs?

p-nitrophenyl assays is a common assay for lipases however MAGs and p-nitrophenyl esters have different head groups. p-nitrophenyl are a more unspecific substrate for fatty acid ester hydrolases and enzymes can have high activities for p-nitrophenyl-FA ester while low activity for MAGs. This might lead to the unexpected results.

My Question basically was if there is a reason why the substrate used was not actual MAGs while measuring released glycerol or fatty acids as read out?

Are all the assays in the manuscript that refer to MAG activity done with p-nitrophenyl-FA-esters? If this is the case it is actually possible the the enzyme has very low MAG activity. That could also explain why intact MAG substrates was found in mass spectroscopy of the crystals.

To address this key question, we have adopted the mono/di/tri acyl glycerol assay published by Rengachari *et al.* (2013) and follow-up papers for the *Tth* enzyme. However, this assay is more complex than the pNP assay we were using previously and very sensitive to experimental conditions a) because its readout is coupled, b) there is residual turnover also in the absence of enzyme requiring careful noise subtraction according to the manufacturer's instructions, and c) there is still ongoing additional residual turnover after *Tth* inactivation by chloroform, requiring consistent timing of termination the reaction and subsequent data analysis. To confirm correct handling of the experimental conditions, we also repeated previous experiments by Rengachari *et al.* (2013) using purified *Bacillus sp.* H257 MAG lipase as positive control. As illustrated in the figure below, we could reproduce previous published data. Our additional data demonstrate that the *Tth* enzyme is highly specific for MAGs as opposed to the corresponding di- and triacyl glyceride esters, with even superior properties in comparison to the control experiments with *Bacillus sp.* H257 MAG lipase. We have not included this into the present version of the manuscript, as we considered this as internal technical control. Should there be a wish to include this into the manuscript it could be added to the supplementary data.

We also found that the *Tth* enzyme activity is about 7-fold higher when using C8 MAG as substrate in comparison to equivalent C8 acyl pNP ester. We reckon that this difference is due to the non-natural pNP leaving group in the latter assay as opposed to measuring MAG hydrolysis. Taking this observation into account, we repeated the assay with C8 MAG for all *Tth* enzyme mutants previously used and have included these data into an additional panel in Figure 4 (panel b, right) and interpret the data in the text. Most of the conclusions arising from this assay are identical or closely related to the previous ones. As we cannot rule out that some of the *Tth* enzyme mutant data using pNP assay may be affected by the non-natural pNP leaving group, we have removed the respective mutant data from

the revised manuscript version. We believe that this is in agreement with the respective point made by referee #2.

Nevertheless, we have retained the *p*NP assay data screening for different monoacyl chain length preferences (Figure 4c), mainly for two reasons: a) because of the superior data precision, as the assay is direct; b) as *p*NP esters are available for acyl compounds of various lengths, this assay provided an opportunity to test the effects of acyl chain length most systematically. As this assay is highly reliable, we also kept the Michaelis-Menten kinetics allowing to dissect catalytic effects (k_{cat}), binding effects (K_M) and estimating catalytic efficiency (k_{cat}/K_M), as illustrated in Figure 4, panels d and e.

By taking the data from both assays together, we are confident of having provided a reasonably complete and quantitative characterization of the *Th* enzyme as specific MAG lipase.

We apologize for a glitch that happened in the previous revised version (not in the original version) where we used the acronym MAG also in the context of this assay by mistake. This has been corrected in this version.

All changes and additions made to address these points are highlighted in the manuscript.

Please clarify what substrate was used (CAS number if necessary), it seems to me the Authors might have tested for the wrong substrate for the conclusions they are trying to make.

Please find the list of CAS numbers listed below, which also include new compounds used in the additional MAG assay.

Abbreviation	Substrate	Distributor	Catalogue No.	CAS number
pNP-C2:0	4-Nitrophenyl acetate	Sigma-Aldrich	N8130	830-03-5
pNP-C4:0	4-Nitrophenyl butyrate	Sigma-Aldrich	N9876	2635-84-9
pNP-C5:0	4-Nitrophenyl valerate	Sigma-Aldrich	N4377	1956-07-6
pNP-C6:0	N-Hexanoic acid 4-nitro-phenyl ester	ABCR	AB139037	956-75-2
pNP-C8:0	4-Nitrophenyl octanoate	Sigma-Aldrich	21742	1956-10-1
pNP-C10:0	4-Nitrophenyl decanoate	Sigma-Aldrich	N0252	1956-09-8
pNP-C12:0	4-Nitrophenyl dodecanoate	Sigma-Aldrich	61716	1956-11-2
pNP-C14:0	Nitrophenyl myristate	Sigma-Aldrich	70124	14617-85-7
pNP-C16:0	p-Nitrophenyl palmitate	Sigma-Aldrich	N2752	1492-30-4
pNP-C18:0	p-Nitrophenyl stearate	Sigma-Aldrich	N3627	14617-86-8
pNP-ferulate	4-Nitrophenyl trans-ferulate	Combi-Blocks	COMH93D5FC7 C	398128-60-4
mono-C8	1-Octanoyl-rac-glycerol	Sigma-Aldrich	M2265	502-54-5

di-C8	1,2-Dioctanoyl-sn-glycerol	Sigma-Aldrich	317505	60514-48-9
tri-C8	Glyceryl trioctanoate	Sigma-Aldrich	T9126	538-23-8
mono-C18(cis9)	1-Oleoyl-rac-glycerol	Sigma-Aldrich	M7765	111-03-5

Reviewer #3 (Remarks to the Author):

The authors have significantly improved the clarity and presentation of the manuscript in the revised version.

The science presented is well supported by experimental data, yet some loose ends remain (which might be the case in any manuscript). The term ester-intermediate is somewhat unusual.

In the revised version the term whenever it was used is replaced by “reaction intermediate”.

Minor comment: Fig 1: D and M are not in the figure, yet described in the figure legend, please correct

Labels have been included.

It is interesting that the authors bring esterification vs hydrolytic reaction of the lipase even more into the focus. As the authors correctly address, it is not entirely clear why this product/substrate has been identified in the active site even upon isolation.

In conclusion, the study on the monoacylglycerol-lipase activity is very nice and comprehensive. It would have been beneficial if more specific ideas on the use of enzymes from Lake Solar were included, ref. 29 and 30 date from 1990 and 1977, respectively. This could widen the interest in the large piece of work broader than the MAGL community.

As we have not been in contact with the authors of these papers that were already published quite a while ago, we believe it would become too speculative to further discuss their original motivation, beyond we have already put into the text extracted from the papers cited.

We also noticed a previously undetected two-fold redundancy of the protein purification protocol in the METHODS part of the manuscript, which in this version has been merged into one under the sub-heading “*Protein expression and purification*”.

REVIEWERS' COMMENTS

Reviewer #3 (Remarks to the Author):

Most discussion points have been addressed.

General remarks:

P11. Reconsider rephrasing: Due to the high level of lipid diversity and dynamics, which is a signature of lipases requiring interfacial activation for substrate turnover ... Is lipid diversity a signature for enzymes for enzymes with interfacial activation?

T: Figure legend S2: monoacylglyceride .. should be monoglyceride or monoacylglycerol:
Similar (and in the entire manuscript) e.g. p6, p10 diacyl and triacyl glycerides should be triacylglycerols;

Some small typos in the manuscript have to be corrected.

Suppl. Table S9, p12 line 367 close to non-methylated Tth MAG lipase (2,3 U mg⁻¹, .. replace , with ‘ ’ for decimal numbers

Figure 4 ... 4b. MAG(C8) or MAG (C18) .. uniform spacing would be optically favorable

Fig 5. Why is *M. tuberculosis* spelled with capital letters, whereas *P. ferrophilus* and others are lower case.

POINT-TO-POINT RESPONSE ON REVIEWER COMMENTS

All replies inserted are in grey-blue both in the manuscript text and point-to-point response.

For convenience, we are also including an updated list of figures and tables.

	Previous version	Revised version	Comments
Figures (Main)	1	1	
	2	2	
	3	3	
	4	4	
	5	5	
		6	Moved from Supplement
Figures (Supplement)	1	1	
	2	2	
	3	3	
	4	4	
	5	5	
	6	6	
	7	7	
	8	8	
	9	9	
	10	10	
	11	11	
	12	12	
	13	13	
	14	15	
	15	Removed	
Tables (Supplement)	1	1	
	2	2	
	3	3	
	4	4	
	5	5	
	6	6	
	7	7	
	8	8	
	9	9	
		10	New
	10	11	
	11	12	

Reviewer #3 (Remarks to the Author):

Most discussion points have been addressed.

General remarks:

P11. Reconsider rephrasing: Due to the high level of lid diversity and dynamics, which is a signature of lipases requiring interfacial activation for substrate turnover ... Is lid diversity a signature for enzymes for enzymes with interfacial activation?

The insert “which is a signature of lipases requiring interfacial activation for substrate turnover” has been deleted.

T: Figure legend S2: monacylglyceride .. should be monoglyceride or monacylglycerol: Similar (and in the entire manuscript) e.g. p6, p10 diacyl and triacyl glycerides should be triacylglycerols;

This has been corrected. Based on this comment, which we very much appreciate, we have reviewed the complete manuscript again, to ensure that the nomenclature used is consistent. Since the acronyms DAG and TAG were also introduced for di- and triacylglycerol, respectively, we have replaced all remaining full words by the respective acronyms. All changes made in the different parts of the manuscript are highlighted.

Some small typos in the manuscript have to be corrected.

Suppl. Table S9, p12 line 367 close to non-methylated Tth MAG lipase (2,3 U mg⁻¹, .. replace , with ‘.’ for decimal numbers

We could not find this typo, however we found others. In Table S9, all decimals were corrected by replacing “,” by “.”. One typo of the same type in Table S6 was also corrected.

Figure 4 ... 4b. MAG(C8) or MAG (C18) .. uniform spacing would be optically favorable Fig 5.

This has been corrected.

Why is m. Tuberculosis spelled with capital letters, whereas P. ferrophilus and others are lower case.

This has been corrected.

Additional note by the authors:

We have noticed that the Solar Lake located on the Sinai peninsula actually belongs Egypt and not Israel (the Sinai peninsula was temporarily occupied by Israel until 1979). We have corrected this by simply stating “...located on the Sanai peninsula...”.